# Provably Mitigating Corruption, Overoptimization, and Verbosity Simultaneously in Offline and Online RLHF/DPO Alignment

## Abstract

Reinforcement learning from human feedback (RLHF) and direct preference optimization (DPO) are important techniques to align large language models (LLM) with human preference. However, the quality of RLHF and DPO training is seriously compromised by *Corrupted* preference, reward *Overoptimization*, and bias towards *Verbosity*. To our knowledge, most existing works tackle only one of these important issues, and the few other works require much computation to estimate multiple reward models and lack theoretical guarantee of generalization ability. In this work, we propose RLHF-**COV** and DPO-**COV** algorithms that can simultaneously mitigate these three issues, in both offline and online settings. This ability is theoretically demonstrated by obtaining length-regularized generalization error rates for our DPO-COV algorithms trained on corrupted data, which match the best-known rates for simpler cases with clean data and without length regularization. Moreover, our DPO-COV algorithm is simple to implement without reward estimation, and is proved to be equivalent to our RLHF-COV algorithm, which directly implies the equivalence between the vanilla RLHF and DPO algorithms.

## 1. Introduction

Reinforcement learning from human feedback (RLHF) has been widely used in robotics (Christiano et al., 2017; Bukharin et al., 2024), autonomous driving (Wang et al., 2024; Cao et al., 2024), large language models (LLM) (Ouyang et al., 2022; Bai et al., 2022b; Rafailov et al., 2023), image and video generation (Wallace et al., 2023; Liang et al., 2024; Liu et al., 2024b), etc. This work will focus on the application of RLHF to LLM alignment which makes LLM more helpful, honest, and harmless (Ouyang et al., 2022; Bai et al., 2022b). LLM alignment has two critical steps. The first step is reward modeling, which estimates the reward model that measures the quality of LLM responses, based on human preference data. The second step is reinforcement learning (RL), which fine-tunes the LLM policy to generate responses with an improved expected value of the learned reward (Ouyang et al., 2022). Direct preference optimization (DPO) (Rafailov et al., 2023) further simplifies the standard RLHF process by directly fine-tuning the optimal policy without reward estimation.

However, the LLM aligned by RLHF and DPO sometimes yields undesirable responses, due to the **corruption**, **overoptimization**, and **verbosity** issues, as introduced below.

**Corruption.** The quality of preference data is essential in RLHF and DPO. However, preference labels given by human may be corrupted due to inexperience, inattention, personal bias, unclear context, and even malicious falsification (Bukharin et al., 2024). For instance, when fine-tuning LLM for automated content moderation on social media, malicious annotators may mislabel harmful contents like misinformation and hate speech as preferable, which misleads the LLM to generate such harmful contents. Therefore, robustness of RLHF and DPO to such corruption is critical, but is tackled by only a few recent works to our knowledge. For example, (Cheng et al., 2024; Mandal et al., 2024; Gao et al., 2024b) use confidence-based data filtering. (Ethayarajh et al., 2024) maximizes the utility function defined based on the prospect theory of human decision making (Tversky and Kahneman, 1992) to filter out noisy data. (Coste et al., 2024; Rame et al., 2024) estimate an ensemble of rewards. The recently proposed robust RLHF and robust DPO approaches in (Bukharin et al., 2024) use noise modeling to automatically select the outliers and the estimated reward provably converges to the true reward.

**Overoptimization.** RLHF and DPO may overoptimize the reward model, yielding LLM responses of high estimated reward but low actual quality (Gao et al., 2023; Casper et al., 2023). Various methods have been proposed to tackle such overoptimization issue (a.k.a. reward hacking). For example, (Gao et al., 2023) uses larger reward model which significantly increases the computational cost of pretraining.

---

[1]Anonymous Institution, Anonymous City, Anonymous Region, Anonymous Country. Correspondence to: Anonymous Author <anon.email@domain.com>.

Preliminary work. Under review by the International Conference on Machine Learning (ICML). Do not distribute.

(Moskovitz et al., 2024) applies constraints to RLHF. The ΦPo method (Azar et al., 2024) optimizes a general preference function. (Eisenstein et al., 2024; Coste et al., 2024; Rame et al., 2024; Fisch et al., 2024; Zhai et al., 2023) use an ensemble of estimated rewards.

An emerging and popular strategy with provable generalization ability to solve overoptimization is to adopt a pessimistic (resp. an optimistic) approach for RLHF and DPO with offline (resp. online) data. Specifically, in the offline setting where only precollected offline preference data is available for training, there are many out-of-distribution samples about which we cannot obtain any information. Therefore, (Zhu et al., 2023; 2024; Liu et al., 2024c; Cen et al., 2024; Ji et al., 2024; Yang et al., 2024; Huang et al., 2024; Xiong et al., 2024; Ye et al., 2024; Fisch et al., 2024) apply pessimistic principle to RLHF or DPO which penalizes LLM from generating such unknown out-of-distribution responses and thus to mitigate overoptimization. Such pessimism principle has also been used in conventional offline RL (Xie et al., 2021; Jin et al., 2021; Rashidinejad et al., 2021; Bai et al., 2022a; Cheng et al., 2022). In contrast, in the online setting where online data can be collected from the up-to-date policy during the training process, optimistic approaches have been used to encourage the collection of unexplored samples to enrich data diversity in RLHF and DPO (Cen et al., 2024; Xie et al., 2024; Zhang et al., 2024; Ye et al., 2024; Xiong et al., 2024) as well as conventional RL (Wei et al., 2017; Zhong and Zhang, 2023; Liu et al., 2023a;b).

**Verbosity.** LLM aligned by vanilla RLHF and DPO is likely to prefer verbose but possibly low-quality responses (Singhal et al., 2023; Chen et al., 2024; Liu et al., 2024a; Dong et al., 2024; Fisch et al., 2024). Multiple methods have been used to tackle verbosity. For example, (Shen et al., 2023; Chen et al., 2024) disentangle length-related reward component. (Guo et al., 2024) instructs the LLM to prefer concise response. (Eisenstein et al., 2024; Fisch et al., 2024; Chakraborty et al., 2024) estimate an ensemble of reward models. (Singhal et al., 2023; Liu et al., 2024a; Dong et al., 2024; Park et al., 2024) use length penalty and similarly (Meng et al., 2024) uses length normalization.

**Our Motivation.** However, to our knowledge, most existing works primarily tackle only one of these three issues (corruption, overoptimization and verbosity). The only method to our knowledge that has been used to tackle all these issues is to estimate an ensemble of reward models (Coste et al., 2024; Fisch et al., 2024; Eisenstein et al., 2024; Rame et al., 2024), which, however, requires much computation and lacks theoretical guarantee of generalization ability. Therefore, we are motivated to ask the following research question.

> **Q:** *Can we design RLHF and DPO algorithms that solve **corruption**, **overoptimization** and **verbosity** simultaneously with simple implementation and theoretical guarantee of generalization ability?*

## 1.1. Our Contributions

We answer the above question affirmatively, by proposing RLHF-**COV** and DPO-**COV** algorithms that simultaneously mitigate *Corruption*, *Overoptimization* and *Verbosity* issues, in both offline and online settings. Specifically, we tackle *Corruption* by noise modeling, tackle *Overoptimization* by pessimistic and optimistic regularizers in the offline and online settings respectively, and tackle *Verbosity* by length regularizer. Our DPO-COV algorithms are almost as simple to implement as the vanilla DPO algorithm without reward model estimation. We prove that our RLHF-COV and DPO-COV are equivalent in the reward-induced policy space in both the offline and online settings. Since our RLHF-COV and DPO-COV algorithms generalize the vanilla RLHF and DPO algorithms respectively, our equivalence result implies that the vanilla RLHF and DPO algorithms are also equivalent. Moreover, we obtain the length-regularized generalization error rates of our DPO-COV algorithms on both offline and online datasets obtained from corrupted preference, and the rates match the existing results in the simple special case with clean dataset and without verbosity regularization. This theoretically demonstrates that our algorithms can simultaneously mitigate the *Corruption*, *Overoptimization* and *Verbosity* issues.

In particular, the effect of noise modeling on the generalization error of learned policy for corrupted data has not been studied to our knowledge, which requires novel proof techniques. The true and estimated noise terms have very different effects on the generalization error, and thus have to be analyzed at different stages. To elaborate, the estimated noise has to be bounded before applying concentration inequality, such that this unbounded estimated noise term can be canceled out by the noise regularizer. In contrast, the true noise has to be bounded after applying the concentration inequality, since the concentration inequality bounds the distance between the true data distribution (with the true noise term) and the estimated data distribution.

## 2. Preliminaries

**Reinforcement learning from human feedback (RLHF).** A large language model (LLM) provides a random language response $a \in \mathcal{X}$ to any given language prompt $x \in \mathcal{X}$ (for example, instruction or question) following the LLM's policy $\pi(\cdot|x)$. Fine-tuning LLM by reinforcement learning from human feedback (RLHF) consists of two critical steps: training reward model and reinforcement learning

(RL) (Ouyang et al., 2022). The reward model is denoted by a function $r(x, a) \in \mathbb{R}$ which measures the quality of the response $a$ given the prompt $x$. To train the reward model, preference data $\mathcal{D} = \{x_i, a_i^w, a_i^\ell\}_{i=1}^N$ of size $N$ is collected where a pair of responses $a_i^w, a_i^\ell$ are generated given each $i$-th prompt $x_i$, and the response $a_i^w$ is more preferable than $a_i^\ell$ (i.e. $a_i^w \succ a_i^\ell$). Such a pairwise preference is widely assumed to follow the Bradley-Terry model (Bradley and Terry, 1952), that is, given prompt $x$, the generated response $a'$ is more desirable than $a$ with the following probability.

$$\mathbb{P}(a' \succ a|x) = \sigma[r^*(x, a') - r^*(x, a)] \tag{1}$$

where $\sigma(x) \stackrel{\text{def}}{=} 1/(1 + e^{-x})$ and $r^*$ is the unknown true reward model. $r^*$ can be estimated by maximum likelihood estimation (MLE), that is, to minimize the following negative log-likelihood function over a certain reward model family $\mathcal{R}$.

$$\min_{r \in \mathcal{R}} -\frac{1}{N} \sum_{i=1}^N \log \sigma[r(x_i, a_i^w) - r(x_i, a_i^\ell)]. \tag{2}$$

Finally, given the estimated reward model $r \in \mathcal{R}$, the optimal policy is obtained by the following optimization problem over the whole policy space $\Pi \stackrel{\text{def}}{=} \{\pi|\pi(\cdot|x) \text{ is a distribution over } \mathcal{A} \text{ for any } x\}$.

$$\max_{\pi \in \Pi} \mathbb{E}_{x \sim \rho, a \sim \pi(\cdot|x)}[r(x, a)]$$
$$- \beta \mathbb{E}_{x \sim \rho} \text{KL}[\pi(\cdot|x)\|\pi_{\text{ref}}(\cdot|x)], \tag{3}$$

where $\rho$ is the prompt distribution, $\pi_{\text{ref}}$ is the reference policy obtained by supervised fine-tuning, and $\text{KL}(p\|q) = \sum_{a \in \mathcal{A}} p(a) \log \frac{p(a)}{q(a)}$ denotes the KL divergence between any pair of response distributions $p, q$ and $\beta > 0$ is the regularizer coefficient which controls the trade-off between generating responses with high expected reward and bounded distance from the reference policy $\pi_{\text{ref}}$.

**Direct preference optimization (DPO).** As introduced above, classical RLHF requires two large-scale optimization problems to learn the reward model $r$ and the optimal policy $\pi$ respectively. DPO (Rafailov et al., 2023) is introduced to remove the reward learning step and thus reducing computation. To elaborate, note that the optimization problem (3) has the following analytical solution.

$$\pi(a|x) = \frac{\pi_{\text{ref}}(a|x)}{Z(x)} \exp\left[\frac{r(x, a)}{\beta}\right], \tag{4}$$

where $Z(x) := \sum_{a' \in \mathcal{A}} \pi_{\text{ref}}(a'|x) \exp[r(x, a')/\beta]$ is the normalization factor. Conversely, given the optimal policy $\pi$, $r(x, a) = \beta \log \frac{\pi(a|x)}{\pi_{\text{ref}}(a|x)}$ is a solution to Eq. (1). Substituting this reward model into the MLE objective (3), (Rafailov et al., 2023) develops the following simple DPO

objective which only requires policy training.

$$\min_{\pi \in \Pi} -\frac{1}{N} \sum_{i=1}^N \log \sigma\Big[\beta \log \frac{\pi(a_i^w|x_i)}{\pi_{\text{ref}}(a_i^w|x_i)}$$
$$- \beta \log \frac{\pi(a_i^\ell|x_i)}{\pi_{\text{ref}}(a_i^\ell|x_i)}\Big]. \tag{5}$$

However, this DPO objective and the aforementioned vanilla RLHF process are prone to suffer from *corrupted* preference, reward *overoptimization*, and bias towards *verbose* response. We will propose our novel variants of RLHF and DPO to solve the three issues simultaneously, for both offline and online settings, in Sections 3 and 4 respectively.

# 3. Our Offline DPO-COV Algorithm

In this section, we will derive our proposed offline RLHF-**COV** objective and offline DPO-**COV** algorithm (Algorithm 1) which simultaneously solve the *Corruption*, *Overoptimization* and *Verbosity* issues, and then obtain the generalization error rates of our offline DPO-COV algorithm.

## 3.1. Our Offline RLHF-COV Objective

**Offline Data from *Corrupted* Preference.**

**Assumption 1.** *The offline data* $\mathcal{D} \stackrel{\text{def}}{=} \{x_i, a_i^{(1)}, a_i^{(-1)}, y_i\}_{i=1}^N = \{x_i, a_i^w, a_i^\ell, y_i\}_{i=1}^N$ *is generated from the following model with corrupted preference.*

$$x_i \sim \rho, \quad a_i^{(-1)}, a_i^{(1)} \sim \pi_b(\cdot|x_i), \tag{6}$$
$$\mathbb{P}(a_i^{(1)} \succ a_i^{(-1)}) = \sigma[r^*(x_i, a_i^{(1)}) - r^*(x_i, a_i^{(-1)}) + \xi_i^*], \tag{7}$$

*where $\pi_b$ denotes the behavior policy and $\xi_i^* \in \mathbb{R}$ denotes the true preference noise for the $i$-th sample. If $a_i^{(1)} \succ a_i^{(-1)}$, assign the label $y_i = 1$ and denote $a_i^w = a_i^{(1)}$ as the more preferable response and $a_i^\ell = a_i^{(-1)}$ as the less preferable response; Otherwise, let $y_i = -1$, $a_i^w = a_i^{(-1)}$, $a_i^\ell = a_i^{(1)}$.*

The above assumption is very similar to that of offline vanilla RLHF and DPO, except that we add noise $\xi_i^*$ to the Bradley-Terry model (1) for each possibly corrupted sample $i$ (Bukharin et al., 2024).

Based on Assumption 1, $\mathbb{P}(y_i|a_i^{(1)}, a_i^{(-1)}) = \sigma[r^*(x_i, a_i^w) - r^*(x_i, a_i^\ell) + y_i\xi_i^*], y_i \in \{-1, 1\}$[1]. Hence, we define a penalized negative log-likelihood function of the labels $\{y_i\}_{i=1}^N$ as follows.

$$\mathcal{L}_{N,\lambda}(r, \xi) \stackrel{\text{def}}{=} -\frac{1}{N} \sum_{i=1}^N \log \sigma[r(x_i, a_i^w) - r(x_i, a_i^\ell) + y_i\xi_i]$$

---

[1] We corrected the mistake in (Bukharin et al., 2024) which uses $\mathbb{P}(y_i|a_i^{(1)}, a_i^{(-1)}) = \sigma[r^*(x_i, a_i^w) - r^*(x_i, a_i^\ell) + \xi_i^*], y_i \in \{-1, 1\}$ that yields $\sum_{y_i \in \{-1,1\}} \mathbb{P}(y_i|a_i^{(1)}, a_i^{(-1)}) \neq 1$.

$$+ \frac{\lambda}{N}\|\xi\|_1, \tag{8}$$

which, compared with the standard non-corrupted negative log-likelihood function (2), adds the estimated preference noise $\xi = [\xi_1, \ldots, \xi_N] \in \mathbb{R}^N$ and the noise regularizer $\|\xi\|_1 = \sum_{i=1}^N |\xi_i|$ with coefficient $\lambda > 0$ to encourage the sparsity of the noise.

**Reward Estimation via Pessimistic MLE to Solve *Overoptimization*.** After collecting offline data, the next step is to learn the reward model $r$. One may consider corrupted MLE objective $\min_{r \in \mathcal{R}, \xi \in \mathbb{R}^N} \mathcal{L}_{N,\lambda}(r, \xi)$ (Bukharin et al., 2024) which generalizes the non-corrupted MLE objective (2). However, this corrupted MLE objective tend to overfit limited offline data (Gao et al., 2023; Zhu et al., 2024; Liu et al., 2024c; Cen et al., 2024; Xiong et al., 2024), producing an inaccurately estimated reward that leads to overoptimization. Therefore, we consider the following pessimistic MLE inspired by (Liu et al., 2024c; Cen et al., 2024; Ji et al., 2024; Yang et al., 2024).

$$\min_{r \in \mathcal{R}, \xi \in \mathbb{R}^N} \left\{ \mathcal{L}_{N,\lambda}(r, \xi) + \eta \max_{\pi \in \Pi} V_\beta(\pi, r) \right\}, \tag{9}$$

where the pessimistic hyperparameter $\eta \geq 0$ and

$$V_\beta(\pi, r) \overset{\text{def}}{=} \mathbb{E}_{x \sim \rho, a \sim \pi(\cdot|x), a' \sim \pi_{\text{base}}(\cdot|x)} \big[ r(x, a) - r(x, a') \big]$$
$$- \beta \mathbb{E}_{x \sim \rho} \text{KL} \big[ \pi(\cdot|x) \big\| \pi_{\text{ref}}(\cdot|x) \big] \tag{10}$$

denotes the relative value of the policy $\pi$ to a certain baseline policy $\pi_{\text{base}}$ given the reward $r$. The regularizer $\max_{\pi \in \Pi} V_\beta(\pi, r)$ in Eq. (9) can be seen as the relative value of the optimal policy, and will help reduce the reward value $r(x, a)$ of any sample $x, a$ with small $\pi_{\text{base}}(a|x)$, so that the optimal policy $\pi(a|x)$ given by Eq. (4) will also be reduced. In other words, such samples $x, a$ are considered pessimistic and are thus discouraged from being generated by the learned policy $\pi$. Hence, the regularizer $\max_{\pi \in \Pi} V_\beta(\pi, r)$ is called the pessimistic regularizer. Furthermore, if we select $\pi_{\text{base}}$ to represent the offline data distribution (see the end of Section 3.2 for the choice of $\pi_{\text{base}}$), then these samples $x, a$ with small $\pi_{\text{base}}(a|x)$ can be seen as out-of-distribution, so that such pessimism on the out-of-distribution samples mitigates the overoptimization issue which often results from overestimation of the reward on low-quality out-of-distribution samples (Liu et al., 2024c).

**Policy Training with Penalized *Verbosity*.** The vanilla RLHF usually yields reward model $r(x, a)$ that has bias towards long and detailed responses. To suppress verbose responses in the policy optimization step $\max_{\pi \in \Pi} V_\beta(\pi, r)$, we can replace the reward model $r(x, a)$ with the proxy reward model $r_\omega(x, a) = r(x, a) - \omega|a|$ where $|a|$ is the length (i.e., number of tokens) of the response $a$ and the hyperparameter $\omega \geq 0$ controls the length penalty strength

(Singhal et al., 2023; Liu et al., 2024a; Dong et al., 2024; Park et al., 2024). In this way, the policy training objective $V_\beta(\pi, r)$ (defined by Eq. (10)) is generalized to the following length-regularized relative value function.

$$V_{\beta,\omega}(\pi, r)$$
$$\overset{\text{def}}{=} \mathbb{E}_{x \sim \rho, a \sim \pi(\cdot|x), a' \sim \pi_{\text{base}}(\cdot|x)} \big[ r(x, a) - \omega|a| - r(x, a')$$
$$+ \omega|a'| \big] - \beta \mathbb{E}_{x \sim \rho} \text{KL} \big[ \pi(\cdot|x) \big\| \pi_{\text{ref}}(\cdot|x) \big]. \tag{11}$$

Replacing $V_\beta(\pi, r)$ with $V_{\beta,\omega}(\pi, r)$ in the pessimistic MLE objective (9), we propose offline RLHF-COV objective below.

**(Offline RLHF-COV):**

$$\min_{r \in \mathcal{R}, \xi \in \mathbb{R}^N} \max_{\pi \in \Pi} \left\{ \mathcal{L}_{N,\lambda}(r, \xi) + \eta V_{\beta,\omega}(\pi, r) \overset{(8),(11)}{=} \right.$$
$$+ \eta \mathbb{E}_{x \sim \rho, a \sim \pi(\cdot|x), a' \sim \pi_{\text{base}}(\cdot|x)}$$
$$\big[ r(x, a) - \omega|a| - r(x, a') + \omega|a'| \big]$$
$$\left. + \frac{1}{N} \sum_{i=1}^N \left\{ \lambda|\xi_i| - \log \sigma[r(x_i, a_i^w) - r(x_i, a_i^\ell) + y_i \xi_i] \right\} \right\}$$
$$- \beta \eta \mathbb{E}_{x \sim \rho} \text{KL} \big[ \pi(\cdot|x) \big\| \pi_{\text{ref}}(\cdot|x) \big]. \tag{12}$$

**Remark:** Our offline RLHF-COV objective above simultaneously tackles the *Corruption*, *Overoptimization* and *Verbosity* issues, via noise modeling, pessimism and length penalty with controllable hyperparameters $\lambda, \eta, \omega$ respectively. Specifically, the length penalty is only added to $V_{\beta,\omega}$ not $\mathcal{L}_{N,\lambda}$, because in the pessimistic MLE we still want to obtain a reward $r$ possibly with length bias, and then verbosity is only suppressed in the policy optimization part $\max_{\pi \in \Pi} V_{\beta,\omega}(\pi, r)$. When $\lambda \geq 1$ and $\eta = \omega = 0$, our offline RLHF-COV objective above reduces to the reward estimation (2) and policy optimization (3) in the vanilla RLHF.

### 3.2. Our Offline DPO-COV Algorithm

The offline RLHF-COV objective (12) involves minimax optimization over three high-dimensional variables $r, \xi, \pi$. As the first step to simplify this objective, we obtain the following proposition.

**Proposition 1.** $(\pi, r, \xi)$ *is the solution to the offline RLHF-COV objective* (12) *if and only if* $\pi = \pi_r \overset{\text{def}}{=} \arg\max_{\pi' \in \Pi} V_{\beta,\omega}(\pi', r)$, $\xi = \xi_r \overset{\text{def}}{=} \arg\min_{\xi \in \mathbb{R}^N} \mathcal{L}_{N,\lambda}(r, \xi)$ *and $r$ is the solution to the following optimization problem.*

$$\min_{r \in \mathcal{R}} [\mathcal{L}_{N,\lambda}(r, \xi_r) + \eta V_{\beta,\omega}(\pi_r, r)]. \tag{13}$$

*In addition, $\pi_r$ and $\xi_{r,i}$ (the $i$-th entry of $\xi_r$) have the following analytical solutions.*

$$\pi_r(a|x) = \frac{\pi_{\text{ref}}(a|x)}{Z_r(x)} \exp \left[ \frac{r(x, a) - \omega|a|}{\beta} \right], \tag{14}$$

$$\xi_{r,i} = y_i I\{\lambda < 1\}$$

$$\left[ \log\left(\frac{1}{\lambda} - 1\right) - r(x_i, a_i^w) + r(x_i, a_i^\ell) \right]_+, \quad (15)$$

where $Z_r(x) \overset{\text{def}}{=} \sum_{a' \in \mathcal{A}} \pi_{\text{ref}}(a'|x) \exp\left[\frac{r(x,a') - \omega|a'|}{\beta}\right]$ is the normalization factor, $I\{\lambda < 1\}$ equals 1 if $\lambda < 1$ and 0 otherwise, and $[u]_+ = \max(u, 0)$ for any $u \in \mathbb{R}$.

The above proposition simplifies the offline RLHF-COV objective (12) into the reward estimation problem (13). Next, we will transform it into our DPO-COV objective of the policy $\pi$. In Eq. (14), given $\pi = \pi_r$, a solution to the reward model $r$ is

$$r^\pi(x, a) \overset{\text{def}}{=} \omega|a| + \beta \log\left[\frac{\pi(a|x)}{\pi_{\text{ref}}(a|x)}\right]. \quad (16)$$

With the above reward $r^\pi$, the corresponding noise can also be parameterized by $\pi$ as $\xi^\pi \overset{\text{def}}{=} \xi_{r^\pi}$, whose $i$-th entry has the following analytical solution based on Eqs. (15) and (16).

$$\xi_i^\pi \overset{\text{def}}{=} \xi_{r^\pi, i} = y_i I\{\lambda < 1\} \left[\log\left(\frac{1}{\lambda} - 1\right) - \omega(|a_i^w| - |a_i^\ell|)\right.$$

$$\left. - \beta \log\left(\frac{\pi(a_i^w|x_i)\pi_{\text{ref}}(a_i^\ell|x_i)}{\pi(a_i^\ell|x_i)\pi_{\text{ref}}(a_i^w|x_i)}\right)\right]_+, \quad (17)$$

Substituting the above $r^\pi$ and $\xi_i^\pi$ into Eq. (13), we propose our DPO-COV objective as follows.

**(Offline DPO-COV):**

$$\min_{\pi \in \Pi_{\mathcal{R}}} \left\{ \mathcal{L}_{N,\lambda}(r^\pi, \xi^\pi) + \eta V_{\beta,\omega}(\pi_{r^\pi}, r^\pi) = \right.$$

$$- \beta\eta \mathbb{E}_{x \sim \rho, a \sim \pi_{\text{base}}(\cdot|x)}\left[\log \pi(a|x)\right]$$

$$+ \frac{1}{N} \sum_{i=1}^N \left[\lambda|\xi_i^\pi| - \log\sigma\left(\omega(|a_i^w| - |a_i^\ell|)\right)\right.$$

$$\left. + \beta \log\frac{\pi(a_i^w|x_i)\pi_{\text{ref}}(a_i^\ell|x_i)}{\pi(a_i^\ell|x_i)\pi_{\text{ref}}(a_i^w|x_i)}\right) + y_i\xi_i^\pi\right] + C_{\text{off}}\right\}, \quad (18)$$

where $C_{\text{off}} \overset{\text{def}}{=} \beta\eta \mathbb{E}_{x \sim \rho, a \sim \pi_{\text{base}}(\cdot|x)}\left[\log \pi_{\text{ref}}(a|x)\right]$ is a constant independent of $\pi$, and we use the reward-induced policy space $\Pi_{\mathcal{R}} \overset{\text{def}}{=} \{\pi_r : r \in \mathcal{R}\}$ since the optimal policy is $\pi_r$ for some reward $r$ based on Proposition 1. Note that such $\Pi_{\mathcal{R}}$ is sufficiently general to admit any parameterized policy $\pi_\theta$ since by defining $\mathcal{R} = \{r^{\pi_\theta} : \theta \in \Theta\}$, we have $\Pi_{\mathcal{R}} = \{\pi_\theta : \theta \in \Theta\}$ based on Lemma 3.

**Remark:** Our proposed offline DPO-COV objective (18) simultaneously tackles *Corruption*, *Overoptimization* and *Verbosity* issues. *Corruption* is modeled by the noise term $\xi^\pi = [\xi_1^\pi, \ldots, \xi_N^\pi]$ which becomes sparser as the hyperparameter $\lambda \geq 0$ increases, and $\xi^\pi = 0$ when $\lambda \geq 1$. *Overoptimization* is tackled by the pessimistic regularizer $-\beta\eta \mathbb{E}_{x \sim \rho, a \sim \pi_{\text{base}}(\cdot|x)}\left[\log \pi(a|x)\right]$ which helps to increase

1: **Inputs:** Hyperparameters $\beta, \eta, \omega, \lambda \geq 0$, offline data $\{x_i, a_i^w, a_i^\ell\}_{i=1}^N$, reference policy $\pi_{\text{ref}}$.
2: **Output:** Obtain policy $\widehat{\pi}$ via the following practical offline DPO-COV objective.

$$\min_{\pi \in \Pi_{\mathcal{R}}} \psi_N(\pi) \overset{\text{def}}{=} \frac{1}{N} \sum_{i=1}^N \left\{ \lambda|\xi_i^\pi| - \beta\eta \log\pi(a_i^w|x) \right.$$

$$- \log\sigma\left[\omega(|a_i^w| - |a_i^\ell|)\right]$$

$$\left. + \beta \log\left(\frac{\pi(a_i^w|x_i)\pi_{\text{ref}}(a_i^\ell|x_i)}{\pi(a_i^\ell|x_i)\pi_{\text{ref}}(a_i^w|x_i)}\right) + y_i\xi_i^\pi\right\}, \quad (19)$$

where $\xi_i^\pi$ is defined by Eq. (17).

$\pi(a|x)$ for in-distribution samples $(x, a)$ well covered by $\pi_{\text{base}}$. *Verbosity* is penalized by the length regularizers $\omega|a_i^w|, \omega|a_i^\ell|$. When $\lambda \geq 1$ and $\eta = \omega = 0$, our above offline DPO-COV objective (18) reduces to the vanilla DPO objective (5).

We formally establish the equivalence between our offline RLHF-COV objective (12) and offline DPO-COV objective (18) in the following Proposition 2, which implies the equivalence between the vanilla RLHF and DPO algorithms as a special case when $\lambda \geq 1$ and $\eta = \omega = 0$.

**Proposition 2.** *A policy $\pi \in \Pi$ is optimal for the offline DPO-COV objective (18) if and only if there exist $r \in \mathcal{R}, \xi \in \mathbb{R}^N$ such that $(\pi, r, \xi)$ is optimal for the offline RLHF-COV objective (12). In this case, $\xi = \xi^\pi$, and for any $x \in \mathcal{X}$, there exists $U_\pi(x) \in \mathbb{R}$ such that $r(x, \cdot) = r^\pi(x, \cdot) + U_\pi(x)$.*

As suggested by (Liu et al., 2024c; Yang et al., 2024) and discussed in Section 3.3, in the DPO-COV objective (18), we can take $\pi_{\text{base}}(\cdot|x)$ as the distribution of the preferable responses $a_i^w$ given $x_i = x$ under Assumption 1, and then adopt the simple stochastic approximation $\mathbb{E}_{x \sim \rho, a \sim \pi_{\text{base}}(\cdot|x)}\left[\log \pi(a|x)\right] \approx \frac{1}{N} \sum_{i=1}^N \log\pi(a_i^w|x_i)$. This yields our fully stochastic offline DPO-COV algorithm as Algorithm 1, which only requires to solve the policy optimization problem that is almost as simple as the vanilla DPO objective (5).

### 3.3. Generalization Analysis of Offline DPO-COV

While the policy $\pi$ is trained from the offline data $\mathcal{D}$, the ultimate goal is to make $\pi$ generalize well to all possible prompts $x \sim \rho$. Specifically, we define the following length-regularized value function which characterizes the generalization ability of the policy $\pi$ as a trade-off among the true reward value $r^*$ (response quality), the length of the

generated response $a$, and the policy's distance to $\pi_{\text{ref}}$.

$$J_{\beta,\omega}(\pi) := \mathbb{E}_{x \sim \rho, a \sim \pi(\cdot|x)} \Big[ r^*(x, a) - \omega|a|$$
$$- \beta \text{KL}\big[\pi(\cdot|x)\big\|\pi_{\text{ref}}(\cdot|x)\big]\Big]. \quad (20)$$

To analyze the generalization error of the policy $\widehat{\pi}$ obtained from Algorithm 1, we make the standard assumptions below.

**Assumption 2** (Realizable and Bounded Reward (Zhu et al., 2023; Zhan et al., 2024; Cen et al., 2024; Ji et al., 2024; Liu et al., 2024c)). *The reward model set $\mathcal{R}$ includes the true reward model $r^*$, that is, $r^* \in \mathcal{R}$. Also, there exists a constant $R \in (0, +\infty)$ such that for any $x \in \mathcal{X}$, $a \in \mathcal{A}$ and $r \in \mathcal{R}$, we have $r(x, a) \in [0, R]$.*

**Assumption 3** (Offline Data Coverage (Zhan et al., 2024; Ji et al., 2024; Liu et al., 2024c)). *There exists a constant $G_{\mathcal{D}} \in (0, +\infty)$ called offline coverage coefficient, such that the choice of the baseline policy $\pi_{\text{base}}$ satisfies the following coverage property for all $r \in \mathcal{R}$.*

$$\mathbb{E}_{x \sim \rho, a \sim \pi_{r^*}(\cdot|x), a' \sim \pi_{\text{base}}(\cdot|x)}$$
$$\big[r^*(x, a) - r^*(x, a') - r(x, a) + r(x, a')\big] \le G_{\mathcal{D}} E_r, \quad (21)$$

*where $E_r \stackrel{\text{def}}{=} \big[\mathbb{E}_{\mathcal{D}}\big|r^*(x_1, a_1^w) - r^*(x_1, a_1^\ell) - r(x_1, a_1^w) + r(x_1, a_1^\ell)\big|^2\big]^{1/2}$ with the offline data sample $x_1, a_1^w, a_1^\ell$ generated via Assumption 1.*

The offline coverage coefficient $G_{\mathcal{D}}$ above describes how well the offline data $\mathcal{D}$ covers the responses from $\pi_{\text{base}}$ and the true optimal policy $\pi_{r^*} \in \arg\max_{\pi \in \Pi} J_{\beta,\omega}(\pi)$. Algorithm 1 takes $\pi_{\text{base}}(\cdot|x)$ as the distribution of the preferable responses $a_i^w$ given $x_i = x$, which is well covered by $\mathcal{D}$.

**Theorem 1.** *Suppose Assumptions 1-3 hold and $\mathcal{R}$ is a convex set. For any $\delta \in (0, 1)$, select hyperparameters $\lambda \in [\sigma(R), 1]$, $\eta = \frac{2\sqrt{\|\xi^*\|_1 + 5\log[|\mathcal{N}_{1/N}(\mathcal{R})|/\delta]}}{\sqrt{N}(3 + e^R)}$. Then, the policy $\widetilde{\pi}$ from the offline DPO-COV objective (18) has the following generalization error rate with probability at least $1 - \delta$.*

$$\max_{\pi \in \Pi} J_{\beta,\omega}(\pi) - J_{\beta,\omega}(\widetilde{\pi})$$
$$\le \frac{(G_{\mathcal{D}}^2 + 1)(3 + e^R)}{\sqrt{N}} \sqrt{\|\xi^*\|_1 + 5\log[|\mathcal{N}_{1/N}(\mathcal{R})|/\delta]}, \quad (22)$$

*where $\mathcal{N}_{1/N}(\mathcal{R})$ is a $(1/N)$-cover of $\mathcal{R}$, that is, for any $r \in \mathcal{R}$, there exists $r^\dagger \in \mathcal{N}_{1/N}(\mathcal{R})$ satisfying $\|r^\dagger - r\|_\infty \le 1/N$.*

**Comparison with Existing Works.** Note that $|\mathcal{N}_{1/N}(\mathcal{R})| \le \mathcal{O}[(RN)^{|\mathcal{X}||\mathcal{A}|}]$ since $\mathcal{R} \subset [0, R]^{|\mathcal{X}||\mathcal{A}|}$ by Assumption 2. Hence, as long as $\|\xi^*\|_1 \le \mathcal{O}[\log(N)]$ (much weaker than Assumption 4.2 of (Bukharin et al.,

2024) that there exist constants $c_0, c_\infty > 0$ such that $\xi^*$ has at most $c_0$ nonzero entries and they range in $[-c_\infty, c_\infty]$), the generalization error rate (22) has the order of $\mathcal{O}[\log(N)/\sqrt{N}]$. This rate matches the existing error rates of the offline pessimistic DPO-type algorithms (Liu et al., 2024c; Cen et al., 2024; Ji et al., 2024) up to logarithm, in the simple case with clean data ($\lambda \ge 1$) and without length regularization ($\omega = 0$). This implies that our offline DPO-COV algorithm provably mitigates **O**veroptimization. In addition, Theorem 1 also for the first time extends to the corrupted data and the length-regularized generalization error, which shows that our Algorithm 1 also mitigates **C**orruption and **V**erbosity. In particular, to mitigate **C**orruption, we use novel techniques below to bound the noise terms in the generalization error of the learned policy, whereas (Bukharin et al., 2024) only analyzes the estimation error of the reward and noise, but not that of the policy.

**Technical Novelty.** The proof logic of Theorem 1 is inspired from that of (Liu et al., 2024c), but our proof requires novel techniques to bound the effects of the true noise $\xi^*$ and estimated noise $\xi^\pi$. To elaborate, the $\xi^\pi$ is analyzed by our proposed Lemma 4, such that the error bound $\sigma(R)|\xi_{r,i}|$ can later be canceled out by the regularizer $-\lambda|\xi_{r,i}|$ when bounding the MLE error in Lemma 8. Next, we bound the distance between the true data distribution under $(r^*, \xi^*)$ and the noiseless data distribution under the estimated $r$ and $\xi = 0$ (see (c) of Eq. (43)) by concentration inequality. Then we bound $\xi^*$ by our proposed Lemma 5 which has a different form from Lemma 4 used for bounding $\xi^\pi$.

## 4. Our Online DPO-COV Algorithm

Compared with offline RLHF and DPO-type algorithms which use precollected offline data, the online algorithms improve the data coverage and the quality of the trained policy (Cen et al., 2024; Dong et al., 2024; Xu et al., 2024; Ye et al., 2024; Guo et al., 2024) at the computation cost of collecting the online preference data in the training process (Zhan et al., 2024; Ji et al., 2024; Huang et al., 2024; Mandal et al., 2024). Therefore, online and offline algorithms have different advantages, so both are important. In this section, we will derive our online RLHF-COV objective and online DPO-COV algorithm, and provide the generalization analysis result of our DPO-COV algorithm.

At each $t$-th iteration of our online algorithm, we use the current policy $\pi_t$ to obtain the $t$-th sample by $x_t \sim \rho$, $a_t^{(-1)} \sim \pi_{\text{ref}}(\cdot|x_t)$, $a_t^{(1)} \sim \pi_t(\cdot|x_t)$, and the label $y_t$ is obtained from a stochastic oracle (such as GPT-4) assumed to follow the corrupted preference model (7). We propose the following online RLHF-COV objective to train the next policy $\pi_{t+1}$ on the online data $\{x_i, a_i^{(-1)}, a_i^{(1)}, y_i\}_{i=1}^t$.

**(Online RLHF-COV):**

$$\pi_{t+1} \in \arg\min_{\pi \in \Pi} \min_{r \in \mathcal{R}, \xi^{(t)} \in \mathbb{R}^t} \left\{ \mathcal{L}_{t,\lambda}(r, \xi^{(t)}) - \eta V_{\beta,\omega}(\pi, r) \right\}$$

$$\stackrel{(8),(11)}{=} \beta\eta \mathbb{E}_{x \sim \rho} \text{KL}\left[\pi(\cdot|x) \big\| \pi_{\text{ref}}(\cdot|x)\right]$$

$$+ \frac{1}{t} \sum_{i=1}^{t} \left\{ \lambda|\xi_i| - \log\sigma[r(x_i, a_i^w) - r(x_i, a_i^\ell) + y_i\xi_i] \right\}$$

$$- \eta \mathbb{E}_{x \sim \rho, a \sim \pi(\cdot|x), a' \sim \pi_{\text{base}}(\cdot|x)}$$

$$\left[r(x, a) + \omega|a| - r(x, a') - \omega|a'|\right], \qquad (23)$$

where $\xi^{(t)} = [\xi_1, \ldots, \xi_t]$ denotes the noise. The above online RLHF-COV objective is similar to the offline RLHF-COV objective (12) with the major difference that they tackle overoptimization in seemingly opposite ways. The offline RLHF-COV objective (12) (i.e., $\min_{r \in \mathcal{R}, \xi \in \mathbb{R}^N}[\mathcal{L}_{N,\lambda}(r, \xi) + \eta \max_{\pi \in \Pi} V_{\beta,\omega}(\pi, r)]$) uses the pessimistic term $+\eta \max_{\pi \in \Pi} V_{\beta,\omega}(\pi, r)$ to discourage LLM from generating out-of-distribution samples. In contrast, inspired by (Cen et al., 2024), our above online RLHF-COV objective (i.e., $\min_{r \in \mathcal{R}, \xi \in \mathbb{R}^N}[\mathcal{L}_{t,\lambda}(r, \xi) - \eta \max_{\pi \in \Pi} V_{\beta,\omega}(\pi, r)]$) uses the sign-flipped optimistic term $-\eta \max_{\pi \in \Pi} V_{\beta,\omega}(\pi, r)$ to encourage LLM to collect out-of-distribution samples to enrich the diversity of the online data to improve policy optimization.

Similar to the offline DPO-COV objective (18), we obtain our online DPO-COV objective as follows.

**(Online DPO-COV):**

$$\pi_{t+1} \in \arg\min_{\pi \in \Pi_{\mathcal{R}}} \left\{ \mathcal{L}_{t,\lambda}(r^\pi, \xi^{\pi,(t)}) - \eta V_{\beta,\omega}(\pi_{r^\pi}, r^\pi) \right\}$$

$$= \beta\eta \mathbb{E}_{x \sim \rho, a \sim \pi_{\text{base}}(\cdot|x)} \left[\log\pi(a|x)\right] + \frac{1}{t} \sum_{i=1}^{t} \left[\lambda|\xi_i^\pi| \right.$$

$$\left. - \log\sigma\left(\omega(|a_i^w| - |a_i^\ell|) + \beta\log\frac{\pi(a_i^w|x_i)\pi_{\text{ref}}(a_i^\ell|x_i)}{\pi(a_i^\ell|x_i)\pi_{\text{ref}}(a_i^w|x_i)}\right) \right.$$

$$\left. + y_i\xi_i^\pi\right] + C_{\text{on}} \Bigg\}, \qquad (24)$$

where $\xi^{\pi,(t)} \stackrel{\text{def}}{=} [\xi_1^\pi, \ldots, \xi_t^\pi]$ is given by Eq. (17) and $C_{\text{on}} = -\beta\eta \mathbb{E}_{x \sim \rho, a \sim \pi_{\text{base}}(\cdot|x)}[\log\pi_{\text{ref}}(a|x)]$ is a constant independent of $\pi$. Similar to Proposition 2, we can show that the online RLHF-COV objective (23) and the online DPO-COV objective (24) are equivalent as follows.

**Proposition 3.** *A policy $\pi \in \Pi$ is optimal for the online DPO-COV objective (24) if and only if there exist $r \in \mathcal{R}, \xi \in \mathbb{R}^N$ such that $(\pi, r, \xi)$ is optimal for the offline RLHF-COV objective (23). In this case, $\xi = \xi^\pi$ and for any $x \in \mathcal{X}$, there exists $U_\pi(x) \in \mathbb{R}$ such that $r(x, \cdot) = r^\pi(x, \cdot) + U_\pi(x)$.*

Inspired by (Xie et al., 2024), we select $\pi_{\text{base}} = \pi_{\text{ref}}$ and use its generated samples $\{a_i^{(-1)}\}_{i=1}^t$ to approximate the

---

**Algorithm 2** Online DPO-COV Algorithm

1: **Inputs:** $\beta, \eta, \omega, \lambda > 0$, reference policy $\pi_{\text{ref}}$, inital policy $\pi_0$.
2: **for** Iterations $t = 1, \ldots, T$ **do**
3:     Generate the $t$-th sample by $x_t \sim \rho$, $a_t^{(-1)} \sim \pi_{\text{ref}}(\cdot|x_t)$, $a_t^{(1)} \sim \pi_t(\cdot|x_t)$, and label $y_t$ from a certain stochastic oracle assumed to follow the corrupted preference model (7).
4:     Obtain $\pi_{t+1}$ by solving the following stochastic online DPO-COV objective (25).

$$\min_{\pi \in \Pi_{\mathcal{R}}} \phi_t(\pi) = \frac{1}{t} \sum_{i=1}^{t} \left\{ \lambda|\xi_i^\pi| + \beta\eta \log\pi(a_i^{(-1)}|x_i) \right.$$

$$- \log\sigma\left[\omega(|a_i^w| - |a_i^\ell|)\right.$$

$$\left. + \beta\log\left(\frac{\pi(a_i^w|x_i)\pi_{\text{ref}}(a_i^\ell|x_i)}{\pi(a_i^\ell|x_i)\pi_{\text{ref}}(a_i^w|x_i)}\right) + y_i\xi_i^\pi\right] \Bigg\}, \quad (25)$$

5: **end for**
6: **Output:** $\pi_{\widehat{T}}$ where $\widehat{T} \sim \text{Uniform}(\{2, 3, \ldots, T, T + 1\})$.

---

expectation in the above online DPO-COV objective. This yields our fully stochastic online DPO-COV algorithm (Algorithm 2), which is also almost as simple to implement as the online vanilla DPO algorithm (Guo et al., 2024) (also Algorithm 2 with $\eta = \omega = 0$ and $\lambda = 1$).

To analyze the generalization error of Algorithm 2, define the following coverability coefficient (Xie et al., 2024), which ensures that there exists at least one policy $\nu \in \Pi_{\mathcal{R}}$ with good coverage over the responses generated by any policy $\pi \in \Pi_{\mathcal{R}}$.

$$G_{\text{on}} \stackrel{\text{def}}{=} \inf_{\nu \in \Pi_{\mathcal{R}}} \sup_{x \in \mathcal{X}, a \in \mathcal{A}, \pi \in \Pi_{\mathcal{R}}} \frac{\pi(a|x)}{\nu(a|x)}. \qquad (26)$$

**Theorem 2.** *Under Assumption 2 and for any $\delta \in (0, 1)$, select hyperparameters $\lambda \in [\sigma(R), 1]$, $\eta = \frac{\sqrt{\log[4T\mathcal{N}_{1/T}(\mathcal{R})/\delta] + \|\xi^*\|_1}}{(3 + e^R)\sqrt{TG_{\text{on}}}}$ where $\xi^* = [\xi_1^*, \ldots, \xi_T^*]$. Then the output policy $\pi_{\widehat{T}}$ of Algorithm 2 satisfies the following generalization error rate with probability at least $1 - \delta$.*

$$\max_{\pi \in \Pi} J_{\beta,\omega}(\pi) - \mathbb{E}\left[J_{\beta,\omega}(\pi_{\widehat{T}})\right] \leq 37(3 + e^R)(\log T)$$

$$\sqrt{\frac{G_{\text{on}}}{T}\left[\log\left(\frac{4T|\mathcal{N}_{1/T}(\mathcal{R})|}{\delta}\right) + \|\xi^*\|_1\right]}. \qquad (27)$$

**Remark:** Theorem 2 above demonstrates that our online DPO-COV algorithm can simultaneously mitigate the *Corruption*, *Overoptimization* and *Verbosity* issues. When $\|\xi^*\|_1 \leq \mathcal{O}(\log T)$, the above generalization error rate is $\widetilde{O}(1/\sqrt{T})$, which also matches the existing results of the

Table 1: Hyperparameter Values and LC-win Rates of Offline DPO-type Algorithms

| Algorithms | $\lambda$ | $\eta$ | $\omega$ | LC-win rates |
|---|---|---|---|---|
| **Our DPO-COV (all 3 components activated)** | 0.7 | 0.0005 | 0.0005 | **7.61%** |
| Robust DPO (***C****orruption* only) | 0.1 | 0 | 0 | 7.04% |
| Pessimistic DPO (***O****veroptimization* only) | 1 | 0.005 | 0 | 5.50% |
| Length-regularized DPO (***V****erbosity* only) | 1 | 0 | 0.0005 | 7.30% |
| Vanilla DPO | 1 | 0 | 0 | 6.29% |
| Reference model $\pi_{\mathrm{ref}}$ | - | - | - | 4.92% |

Table 2: Experimental Results on Math and Reasoning

| Model | GSM8K | ARC (Easy) | ARC (Challenge) |
|---|---|---|---|
| **Our DPO-COV** | **46.78** | **72.52** | **49.32** |
| Robust DPO | 46.25 | 72.14 | 47.35 |
| Pessimistic DPO | 45.19 | 72.14 | 46.16 |
| Length-reg DPO | 44.50 | 72.31 | 46.16 |
| Vanilla DPO | 45.26 | 71.89 | 46.50 |
| Reference Model | 42.38 | 71.72 | 45.14 |

online optimistic DPO-type algorithms (Xie et al., 2024; Cen et al., 2024) up to logarithm.

**Technical Novelty.** Similar to the proof of Theorem 1, we also use the novel bounds on the effect of the estimated and true noise terms, which are obtained in Lemmas 5 and 4 respectively.

## 5. Experiments on Offline Data

In this section, we will compare the following offline DPO-type algorithms on offline datasets. The experiments to compare online DPO-type algorithms on online datasets are shown in Appendix A.

1. Our offline DPO-**COV** algorithm with three modules activated (***C****orruption*, ***O****veroptimization*, ***V****erbosity*): This is Algorithm 1 with $\eta, \omega > 0$ and $\lambda \in (0, 1)$.

2. Offline robust DPO algorithm (Bukharin et al., 2024): This is a special case of Algorithm 1 with $\eta = \omega = 0$ and $\lambda \in (0, 1)$, which only tackles ***C****orruption*.

3. Offline pessimistic DPO algorithm (Liu et al., 2024c): This is a special case of Algorithm 1 with $\eta > 0, \omega = 0$ and $\lambda = 1$, which only tackles ***O****veroptimization*.

4. Offline length regularized DPO algorithm (Park et al., 2024): This is a special case of Algorithm 1 with $\eta = 0$, $\omega > 0$ and $\lambda = 1$, which only tackles ***V****erbosity*.

5. Offline vanilla DPO (Rafailov et al., 2023): Algorithm 1 with $\eta = \omega = 0$ and $\lambda = 1$.

### 5.1. Experiment on the Argilla Data

We select the preference dataset $\mathcal{D}$ to be Argilla-DPO-Mix-7K (Argill, 2024), and $\pi_{\mathrm{ref}}$ to be zephyr-7b-gemma-sft-v0.1 (HuggingFaceH4, 2024), which is a fine-tuned version of gemma-7b on the Deita dataset (Wang et al., 2023). Then we apply LoRA (Hu et al., 2021) and two epochs of the AdamW optimizer (Loshchilov and Hutter, 2017) with learning rate $5 \times 10^{-7}$ to the objective (19). For each algorithm, we fix $\beta = 0.05$ and perform grid search on the other hyperparameters over a holdout validation set of the preference dataset. We compare the Length-Control win rates (a.k.a. LC-win rates, defined in AlpacaEval 2.0 (Dubois et al., 2024)) of $\pi_{\mathrm{ref}}$ and that of the models obtained by the above algorithms against the model GPT-4 Preview (11/06) (OpenAI, 2024). We summarize the LC-win rates and the hyperparameter values in Table 1, which indicates that our offline DPO-COV algorithm with all three components activated achieves the highest LC win rates. Therefore, it is important to tackle the ***C****orruption*, ***O****veroptimization* and ***V****erbosity* issues simultaneously.

### 5.2. Experiment on Math and Reasoning

We also compare our Algorithm 1 with other offline DPO variants over math and reasoning tasks: Grade School Math 8K (GSM8K) and AI2 Reasoning Challenge (ARC) tasks. We run the benchmark test with (Gao et al., 2024a) and report the accuracies in Table 2. The model hyper-parameters are the same as in Table 1. The results shown in Table 2 indicate that our DPO-COV algorithm outperforms the other variants also on the math and reasoning tasks.

## 6. Conclusion

We proposed RLHF-COV and DPO-COV algorithms that simultaneously mitigate the ***C****orruption*, ***O****veroptimization* and ***V****erbosity* issues, in both offline and online settings. This ability is theoretically proved by length-regularized generalization analysis on corrupted data. In addition, we proved the equivalence of our proposed RLHF-COV and DPO-COV algorithms. A future direction is to extend this work to account for various preferences among diverse human groups (Ramesh et al., 2024; Chakraborty et al., 2024).

## Impact Statement

This paper presents work whose goal is to advance the field of Machine Learning. There are many potential societal consequences of our work, none which we feel must be specifically highlighted here.

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

# Appendix

## Table of Contents

## A. Experiment on Online Data

Similar to the offline experiments in Section 5, we compare important special cases of Algorithm 2, including our online DPO-COV with all 3 components activated, the online variant of the robust DPO algorithm (Bukharin et al., 2024), online optimistic DPO algorithm (named XPO in (Xie et al., 2024)), online length regularized DPO algorithm (Liu et al., 2024a) and online vanilla DPO algorithm (using DPO objective in (Guo et al., 2024)). We use zephyr-7b-gemma-sft-v0.1 (HuggingFaceH4, 2024) as the reference model $\pi_{\mathrm{ref}}$ and the initial model $\pi_0$. Each algorithm is trained with $\beta = 0.05$ and $T = 3$ iterations. In each iteration, we generate the online labels $y_t$ from pair-preference-model-LLaMA3-8B (RLHFlow, 2024), and combine the online data with 50% of the preference dataset of Argilla-DPO-Mix-7K (Argill, 2024). Then we apply LoRA (Hu et al., 2021) and two epochs of the AdamW optimizer (Loshchilov and Hutter, 2017) with stepsize $5 \times 10^{-7}$ to the objective (25). On AlpacaEval 2.0 (Dubois et al., 2024), we compare the LC-win rates of $\pi_{\mathrm{ref}}$ and that of the models obtained by the above algorithms against the model GPT-4 Preview (11/06) (OpenAI, 2024). Again, the results in Table 3 indicate that our online DPO-COV algorithm with all three components activated achieves the highest length-control win rates. Therefore, it is important to tackle the *Corruption*, *Overoptimization* and *Verbosity* issues simultaneously.

Table 3: Hyperparameter Values and LC-win Rates of Online DPO-type Algorithms

| Algorithms | $\lambda$ | $\eta$ | $\omega$ | LC-win rates |
|---|---|---|---|---|
| **Our DPO-COV (all 3 components activated)** | **0.7** | **0.0005** | **0.0005** | **7.87%** |
| Robust DPO (*Corruption* only) | 0.1 | 0 | 0 | 7.03% |
| Optimistic DPO (*Overoptimization* only) | 1 | 0.005 | 0 | 6.23% |
| Length-regularized DPO (*Verbosity* only) | 1 | 0 | 0.0005 | 6.19% |
| Vanilla DPO | 1 | 0 | 0 | 6.58% |
| Reference model $\pi_{\mathrm{ref}}$ | - | - | - | 4.92% |

## B. Supporting Lemmas

**Lemma 1.** *For any $A \in (0, \infty)$ and $z_1, z_2 \in [-R, R]$, the following inequality holds.*

$$\frac{|z_1 - z_2|}{3 + e^R} \leq |\sigma(z_1) - \sigma(z_2)| \leq \frac{1}{4}|z_1 - z_2|. \tag{28}$$

**Remark:** Our bound (28) is strictly tighter than $\frac{|z_1-z_2|}{(1+e^R)^2} \leq |\sigma(z_1) - \sigma(z_2)| \leq |z_1 - z_2|$ obtained in Lemma A.2 of (Liu et al., 2024c).

*Proof.* Denote $z_{\min} = \min(z_1, z_2)$ and $z_{\max} = \max(z_1, z_2)$. Then we have

$$|z_1 - z_2| = z_{\max} - z_{\min},$$

$$|\sigma(z_1) - \sigma(z_2)| = \sigma(z_{\max}) - \sigma(z_{\min}) = \int_{z_{\min}}^{z_{\max}} \sigma'(z)dz.$$

Hence, it suffices to prove that $\sigma'(v) \in \left[\frac{1}{3+e^R}, \frac{1}{4}\right]$ for any $v \in [z_{\min}, z_{\max}] \subset [-R, R]$. Note that for any $v \in [z_{\min}, z_{\max}] \subset [-R, R]$, $\sigma(v) \in [\sigma(-R), \sigma(R)] = [1 - \sigma(R), \sigma(R)]$. Hence, we conclude the proof by the following two bounds.

$$\sigma'(v) = \sigma(v)[1 - \sigma(v)] = \frac{1}{4} - \left[\sigma(v) - \frac{1}{2}\right]^2 \leq \frac{1}{4}.$$

$$
\begin{aligned}
\sigma'(v) &= \frac{1}{4} - \left[\sigma(v) - \frac{1}{2}\right]^2 \\
&\geq \frac{1}{4} - \left[\sigma(R) - \frac{1}{2}\right]^2 \\
&= \sigma(R)[1 - \sigma(R)] \\
&= \frac{1}{1 + e^R} \frac{e^R}{1 + e^R} \\
&= \frac{1}{(1 + e^R)(1 + e^{-R})} \\
&= \frac{1}{2 + e^R + e^{-R}} \geq \frac{1}{3 + e^R}.
\end{aligned}
$$

$\square$

**Lemma 2.** *For any $x \in \mathcal{X}$, $a_0, a_1 \in \mathcal{A}$ and $r \in \mathcal{R}$, the following equality holds*

$$r^{\pi_r}(x, a_1) - r^{\pi_r}(x, a_0) = r(x, a_1) - r(x, a_0), \tag{29}$$

*where $\pi_r$ and $r^\pi$ are defined by Eqs. (14) and (16) respectively. Furthermore, under Assumption 2, both sides of the above Eq. (29) range in $[-R, R]$.*

*Proof.*

$$
\begin{aligned}
&r^{\pi_r}(x, a_1) - r^{\pi_r}(x, a_0) \\
&\overset{(a)}{=} \omega(|a_1| - |a_0|) + \beta \log\left(\frac{\pi_r(a_1|x)\pi_{\text{ref}}(a_0|x)}{\pi_r(a_0|x)\pi_{\text{ref}}(a_1|x)}\right) \\
&\overset{(b)}{=} r(x, a_1) - r(x, a_0),
\end{aligned}
$$

where (a) uses Eq. (16) and (b) uses Eq. (14).

Furthermore, under Assumption 2, $r(x, a_0), r(x, a_1) \in [0, R]$, so

$$r^{\pi_r}(x, a_1) - r^{\pi_r}(x, a_0) = r(x, a_1) - r(x, a_0) \in [-R, R].$$

$\square$

**Lemma 3.** *Any policy $\pi \in \Pi$ satisfies $\pi = \pi_{r^\pi}$ where $\pi_r$ and $r^\pi$ are defined by Eqs. (14) and (16) respectively. Furthermore, under Assumption 2, any $\pi \in \Pi_{\mathcal{R}} \overset{\text{def}}{=} \{\pi_r : r \in \mathcal{R}\}$ satisfies $|r^\pi(x, a_1) - r^\pi(x, a_0)| \leq R$ for any $x \in \mathcal{X}$, $a_0, a_1 \in \mathcal{A}$.*

*Proof.* Eq. (16) implies that for any $x \in \mathcal{X}$ and $a \in \mathcal{A}$, we have

$$\pi_{\text{ref}}(a|x) \exp\left[\frac{r^\pi(x,a) - \omega|a|}{\beta}\right] = \pi(a|x). \tag{30}$$

Hence,

$$Z_{r^\pi}(x) = \sum_{a \in \mathcal{A}} \pi_{\text{ref}}(a|x) \exp\left[\frac{r^\pi(x,a) - \omega|a|}{\beta}\right] = \sum_{a \in \mathcal{A}} \pi(a|x) = 1. \tag{31}$$

Therefore, $\pi = \pi_{r^\pi}$ can be proved as follows.

$$\pi_{r^\pi}(a|x) \stackrel{(a)}{=} \frac{\pi_{\text{ref}}(a|x)}{Z_{r^\pi}(x)} \exp\left[\frac{r^\pi(x,a) - \omega|a|}{\beta}\right] \stackrel{(b)}{=} \pi(a|x),$$

where (a) uses Eq. (14) and (b) uses Eqs. (30) and (31).

When $\pi \in \Pi_\mathcal{R} \stackrel{\text{def}}{=} \{\pi_r : r \in \mathcal{R}\}$, there exists $r \in \mathcal{R}$ such that $\pi = \pi_r$. Hence,

$$|r^\pi(x,a_1) - r^\pi(x,a_0)| \stackrel{(a)}{=} |r^{\pi_r}(x,a_1) - r^{\pi_r}(x,a_0)| \stackrel{(b)}{=} |r(x,a_1) - r(x,a_0)| \stackrel{(c)}{\leq} R,$$

where (a) uses $\pi = \pi_r$, (b) uses Eq. (29) and (c) uses Assumption 2. $\qquad\square$

**Lemma 4.** *Under Assumption 2, for any $r \in \mathcal{R}$ and $\xi_{r,i}$ defined by Eq. (15), the following inequality holds.*

$$\log \sigma[r(x_i, a_i^w) - r(x_i, a_i^\ell) + y_i\xi_{r,i}] \leq \log \sigma[r(x_i, a_i^w) - r(x_i, a_i^\ell)] + \sigma(R)|\xi_{r,i}|. \tag{32}$$

*For any $\pi \in \Pi_\mathcal{R} \stackrel{\text{def}}{=} \{\pi_r : r \in \mathcal{R}\}$ and $\xi_i^\pi$ defined by Eq. (17), the following inequality holds.*

$$\log \sigma[r^\pi(x_i, a_i^w) - r^\pi(x_i, a_i^\ell) + y_i\xi_i^\pi] \leq \log \sigma[r^\pi(x_i, a_i^w) - r^\pi(x_i, a_i^\ell)] + \sigma(R)|\xi_i^\pi|. \tag{33}$$

*Proof.* $y_i\xi_{r,i} \geq 0$ by Eq. (15) since $y_i \in \{-1, 1\}$. Then Eq. (32) follows from $\frac{d}{dv}[\log \sigma(v)] = \sigma(-v) \leq \sigma(R)$ for any $v \in [r(x_i, a_i^w) - r(x_i, a_i^\ell), r(x_i, a_i^w) - r(x_i, a_i^\ell) + y_i\xi_{r,i}] \subseteq [-R, +\infty)$ where $\subset$ is implied by Assumption 2.

Similarly, $y_i\xi_i^\pi \geq 0$ by Eq. (17) since $y_i \in \{-1, 1\}$. Then Eq. (33) follows from $\frac{d}{dv}[\log \sigma(v)] = \sigma(-v) \leq \sigma(R)$ for any $v \in [r^\pi(x_i, a_i^w) - r^\pi(x_i, a_i^\ell), r^\pi(x_i, a_i^w) - r^\pi(x_i, a_i^\ell) + y_i\xi_i^\pi] \subseteq [-R, +\infty)$ where $\subset$ is implied by Lemma 3. $\qquad\square$

**Lemma 5.** *For any $\xi_i \in \mathbb{R}$ and reward models $r, r' : \mathcal{X} \times \mathcal{A} \to \mathbb{R}$, we have*

$$\left\{\sigma[r'(x_i, a_i^w) - r'(x_i, a_i^\ell) + y_i\xi_i] - \sigma[r(x_i, a_i^w) - r(x_i, a_i^\ell)]\right\}^2$$
$$\geq \left\{\sigma[r'(x_i, a_i^w) - r'(x_i, a_i^\ell)] - \sigma[r(x_i, a_i^w) - r(x_i, a_i^\ell)]\right\}^2 - \frac{1}{2}|\xi_i^*|. \tag{34}$$

*Proof.* Denote $A_i' = r'(x_i, a_i^w) - r'(x_i, a_i^\ell)$ and $A_i = r(x_i, a_i^w) - r(x_i, a_i^\ell)$. Define the following function.

$$f(u) = \left[\sigma(A_i' + u) - \sigma(A_i)\right]^2. \tag{35}$$

Note that the range of the sigmoid function $\sigma$ is $(0, 1)$. Hence, for any $u \in \mathbb{R}$,

$$\frac{d}{du}f(u) = 2\sigma(A_i' + u)\left[1 - \sigma(A_i' + u)\right]\left[\sigma(A_i' + u) - \sigma(A_i)\right] \in \left(-\frac{1}{2}, \frac{1}{2}\right). \tag{36}$$

Therefore,

$$f(0) - f(y_i\xi_i) \leq |f(y_i\xi_i) - f(0)| \leq \frac{1}{2}|y_i\xi_i| = \frac{1}{2}|\xi_i|,$$

which implies Eq. (34). $\qquad\square$

**Lemma 6.** *For any $x \in \mathcal{X}$, $a \in \mathcal{A}$ and $r, r' \in \mathcal{R}$, the policies $\pi_r$, $\pi_{r'}$ defined by the analytical solution* (14) *satisfy*

$$\left| \log \frac{\pi_{r'}(a|x)}{\pi_r(a|x)} \right| \leq \frac{2\|r' - r\|_\infty}{\beta}, \tag{37}$$

*where $\|r' - r\|_\infty = \sup_{x \in \mathcal{X}, a \in \mathcal{A}} |r'(x, a) - r(x, a)|$.*

*Proof.* Note that for any $x \in \mathcal{X}$, $a' \in \mathcal{A}$ and $r, r' \in \mathcal{R}$, we have

$$\frac{\pi_{\text{ref}}(a'|x) \exp\left[\frac{r'(x,a')-\omega|a'|}{\beta}\right]}{\pi_{\text{ref}}(a'|x) \exp\left[\frac{r(x,a')-\omega|a'|}{\beta}\right]} = \exp\left[\frac{r'(x, a') - r(x, a')}{\beta}\right]$$

$$\in \left[ \exp(-\|r' - r\|_\infty/\beta), \exp(\|r' - r\|_\infty/\beta) \right].$$

Therefore,

$$\frac{Z_{r'}(x)}{Z_r(x)} = \frac{\sum_{a' \in \mathcal{A}} \pi_{\text{ref}}(a'|x) \exp\left[\frac{r'(x,a')-\omega|a'|}{\beta}\right]}{\sum_{a' \in \mathcal{A}} \pi_{\text{ref}}(a'|x) \exp\left[\frac{r(x,a')-\omega|a'|}{\beta}\right]}$$

$$\in \left[ \exp(-\|r' - r\|_\infty/\beta), \exp(\|r' - r\|_\infty/\beta) \right].$$

As a result,

$$\frac{\pi_{r'}(a|x)}{\pi_r(a|x)} = \left( \frac{Z_{r'}(x)}{Z_r(x)} \right)^{-1} \frac{\pi_{\text{ref}}(a'|x) \exp\left[\frac{r'(x,a')-\omega|a'|}{\beta}\right]}{\pi_{\text{ref}}(a'|x) \exp\left[\frac{r(x,a')-\omega|a'|}{\beta}\right]}$$

$$\in \left[ \exp(-2\|r' - r\|_\infty/\beta), \exp(2\|r' - r\|_\infty/\beta) \right] \tag{38}$$

which directly implies Eq. (37). $\qquad\square$

We slightly adjust Theorem 13.2 of (Zhang, 2023) as follows, by using filtration $\mathcal{F}_t = \emptyset$ (so the conditional expectation becomes the total expectation), replacing $-\xi_i$ with $Z_i$, and negating the small probability event.

**Lemma 7.** *Consider random variables $\{Z_i\}_{i=0}^N$. For any $\delta \in (0, 1)$ and $\lambda' > 0$, the following inequality holds simultaneously for all $n = 1, 2, \ldots, N$ with probability at least $1 - \delta$.*

$$\sum_{i=1}^n Z_i \leq \frac{\log(1/\delta)}{\lambda'} + \frac{1}{\lambda'} \sum_{i=1}^n \log \mathbb{E}[\exp(\lambda' Z_i)].$$

**Lemma 8.** *Fix $\epsilon > 0$, $\lambda \in [\sigma(R), 1]$ and $\delta \in (0, 1)$. Under Assumption 1, the following bound holds for any $r \in \mathcal{R}$ and $\xi_r = [\xi_{r,1}, \ldots, \xi_{r,N}] \in \mathbb{R}^N$ (given by Eq. (15)) simultaneously with probability at least $1 - \delta$.*

$$\mathcal{L}_{N,\lambda}(r^*, \xi^*) - \mathcal{L}_{N,\lambda}(r, \xi_r) \leq \frac{2}{N}\left[ \|\xi^*\|_1 + \log\left(\frac{|\mathcal{N}_\epsilon(\mathcal{R})|}{\delta}\right) \right] - \frac{E_r^2}{2(3 + e^R)^2} + 7\epsilon, \tag{39}$$

*where $E_r := \sqrt{\mathbb{E}_{\mathcal{D}}\left| r^*(x_1, a_1^w) - r^*(x_1, a_1^\ell) - r(x_1, a_1^w) + r(x_1, a_1^\ell) \right|^2}$ and $\mathcal{N}_\epsilon(\mathcal{R})$ is a finite $\epsilon$-cover of $\mathcal{R}$, that is, for any $r \in \mathcal{R}$, there exists $r^\dagger \in \mathcal{N}_\epsilon(\mathcal{R})$ satisfying $\|r^\dagger - r\|_\infty \leq \epsilon$.*

*Proof.* Based on Assumption 1, given $(x_i, a_i^{(1)}, a_i^{(-1)})$, the target label $y \in \{-1, 1\}$ as well as the underlying reward $r$ and noise $\xi_i$, the event $y_i = y$ occurs with the following probability.

$$p_{r,\xi_i}(y|x_i, a_i^{(1)}, a_i^{(-1)}) = \begin{cases} \sigma[r(x_i, a_i^{(1)}) - r(x_i, a_i^{(-1)}) + \xi_i], & y = 1 \\ \sigma[r(x_i, a_i^{(-1)}) - r(x_i, a_i^{(1)}) - \xi_i], & y = -1. \end{cases} \tag{40}$$

By merging the two cases above, we have

$$p_{r,\xi_i}(y_i|x_i, a_i^{(1)}, a_i^{(-1)}) = \sigma[r(x_i, a_i^w) - r(x_i, a_i^\ell) + y_i\xi_i]. \tag{41}$$

Define the following random variables for $r \in \mathcal{R}$ and $i = 1, \ldots, N$.

$$Z_i(r) = \frac{1}{2} \log \frac{\sigma[r(x_i, a_i^w) - r(x_i, a_i^\ell)]}{\sigma[r^*(x_i, a_i^w) - r^*(x_i, a_i^\ell) + y_i \xi_i^*]} = \frac{1}{2} \log \frac{p_{r,0}(y_i | x_i, a_i^{(1)}, a_i^{(-1)})}{p_{r^*, \xi_i^*}(y_i | x_i, a_i^{(1)}, a_i^{(-1)})}. \tag{42}$$

Then the following inequality holds for finitely many $r \in \mathcal{N}_\epsilon(\mathcal{R})$ simultaneously with probability at least $1 - \delta$.

$$\mathcal{L}_{N,\lambda}(r^*, \xi^*) - \mathcal{L}_{N,\lambda}(r, \xi_r)$$

$$= \frac{1}{N} \sum_{i=1}^{N} \left\{ \log \sigma[r(x_i, a_i^w) - r(x_i, a_i^\ell) + y_i \xi_{r,i}] - \log \sigma[r^*(x_i, a_i^w) - r^*(x_i, a_i^\ell) + y_i \xi_i^*] + \lambda(|\xi_i^*| - |\xi_{r,i}|) \right\}$$

$$\overset{(a)}{\leq} \frac{1}{N} \sum_{i=1}^{N} \Big\{ \log \sigma[r(x_i, a_i^w) - r(x_i, a_i^\ell)] + \sigma(R)|\xi_{r,i}| - \log \sigma[r^*(x_i, a_i^w) - r^*(x_i, a_i^\ell) + y_i \xi_i^*]$$

$$+ \lambda(|\xi_i^*| - |\xi_{r,i}|) \Big\}$$

$$\overset{(b)}{\leq} \frac{1}{N} \sum_{i=1}^{N} \left[ |\xi_i^*| + 2 Z_i(r) \right]$$

$$\overset{(c)}{\leq} \frac{1}{N} \sum_{i=1}^{N} \left\{ |\xi_i^*| + 2 \log \mathbb{E}_{\mathcal{D}} \left[ \exp[Z_i(r)] \right] \right\} + \frac{2}{N} \log \left( \frac{|\mathcal{N}_\epsilon(\mathcal{R})|}{\delta} \right)$$

$$\overset{(d)}{=} \frac{2}{N} \sum_{i=1}^{N} \log \mathbb{E}_{\mathcal{D}} \left\{ \mathbb{E}_{y_i \sim p_{r^*, \xi_i^*}(\cdot | x_i, a_i^{(1)}, a_i^{(-1)})} \left[ \sqrt{\frac{p_{r,0}(y_i | x_i, a_i^{(1)}, a_i^{(-1)})}{p_{r^*, \xi_i^*}(y_i | x_i, a_i^{(1)}, a_i^{(-1)})}} \middle| x_i, a_i^{(1)}, a_i^{(-1)} \right] \right\}$$

$$+ \frac{1}{N} \left[ \|\xi^*\|_1 + 2 \log \left( \frac{|\mathcal{N}_\epsilon(\mathcal{R})|}{\delta} \right) \right]$$

$$\overset{(e)}{\leq} \frac{2}{N} \sum_{i=1}^{N} \mathbb{E}_{\mathcal{D}} \left[ \sum_{y \in \{-1,1\}} \sqrt{p_{r,0}(y | x_i, a_i^{(1)}, a_i^{(-1)}) p_{r^*, \xi_i^*}(y | x_i, a_i^{(1)}, a_i^{(-1)})} - 1 \right]$$

$$+ \frac{1}{N} \left[ \|\xi^*\|_1 + 2 \log \left( \frac{|\mathcal{N}_\epsilon(\mathcal{R})|}{\delta} \right) \right]$$

$$= -\frac{1}{N} \sum_{i=1}^{N} \mathbb{E}_{\mathcal{D}} \left[ \sum_{y \in \{-1,1\}} \left| \sqrt{p_{r,0}(y | x_i, a_i^{(1)}, a_i^{(-1)})} - \sqrt{p_{r^*, \xi_i^*}(y | x_i, a_i^{(1)}, a_i^{(-1)})} \right|^2 \right]$$

$$+ \frac{1}{N} \left[ \|\xi^*\|_1 + 2 \log \left( \frac{|\mathcal{N}_\epsilon(\mathcal{R})|}{\delta} \right) \right]$$

$$\overset{(f)}{\leq} -\frac{1}{4N} \sum_{i=1}^{N} \mathbb{E}_{\mathcal{D}} \left[ \sum_{y \in \{-1,1\}} \left| p_{r,0}(y | x_i, a_i^{(1)}, a_i^{(-1)}) - p_{r^*, \xi_i^*}(y | x_i, a_i^{(1)}, a_i^{(-1)}) \right|^2 \right]$$

$$+ \frac{1}{N} \left[ \|\xi^*\|_1 + 2 \log \left( \frac{|\mathcal{N}_\epsilon(\mathcal{R})|}{\delta} \right) \right]$$

$$\overset{(g)}{=} -\frac{1}{2N} \sum_{i=1}^{N} \mathbb{E}_{\mathcal{D}} \left\{ \sigma[r^*(x_i, a_i^w) - r^*(x_i, a_i^\ell) + y_i \xi_i^*] - \sigma[r(x_i, a_i^w) - r(x_i, a_i^\ell)] \right\}^2$$

$$+ \frac{1}{N} \left[ \|\xi^*\|_1 + 2 \log \left( \frac{|\mathcal{N}_\epsilon(\mathcal{R})|}{\delta} \right) \right]$$

$$\overset{(h)}{\leq} -\frac{1}{2N} \sum_{i=1}^{N} \left\{ \mathbb{E}_{\mathcal{D}} \left\{ \sigma[r^*(x_i, a_i^w) - r^*(x_i, a_i^\ell)] - \sigma[r(x_i, a_i^w) - r(x_i, a_i^\ell)] \right\}^2 - \frac{1}{2} |\xi_i^*| \right\}$$

$$+ \frac{1}{N} \left[ \|\xi^*\|_1 + 2 \log \left( \frac{|\mathcal{N}_\epsilon(\mathcal{R})|}{\delta} \right) \right]$$

$$\overset{(i)}{\leq} -\frac{1}{2(3 + e^R)^2} \mathbb{E}_{\mathcal{D}} \left| r^*(x_1, a_1^w) - r^*(x_1, a_1^\ell) - r(x_1, a_1^w) + r(x_1, a_1^\ell) \right|^2$$

$$+ \frac{2}{N}\Big[\|\xi^*\|_1 + \log\Big(\frac{|\mathcal{N}_\epsilon(\mathcal{R})|}{\delta}\Big)\Big]$$

$$\stackrel{(j)}{=} \frac{2}{N}\Big[\|\xi^*\|_1 + \log\Big(\frac{|\mathcal{N}_\epsilon(\mathcal{R})|}{\delta}\Big)\Big] - \frac{E_r^2}{2(3+e^R)^2}, \tag{43}$$

where (a) uses Eq. (32) from Lemma 4, (b) uses Eq. (42) and $\sigma(R) \leq \lambda \leq 1$, (c) denotes $\mathbb{E}_\mathcal{D}$ as the expectation under Assumption 1 and (c) holds for finitely many $r \in \mathcal{N}_\epsilon(\mathcal{R})$ simultaneously with probability at least $1 - \delta$ (by Lemma 7 with $\lambda' = 1$), (d) uses Eq. (42) and Assumption 1, (e) uses $\log v \leq v - 1$ for any $v > 0$, (f) uses Lemma 12.2 of (Harsha, 2011), (g) uses Eq. (41), (h) uses Lemma 5, (i) uses Lemma 1 as well as the fact that the $N$ samples $\{x_i, a_i^w, a_i^\ell\}_{i=1}^N$ are i.i.d., (j) denotes $E_r := \sqrt{\mathbb{E}_\mathcal{D}\big|r^*(x_1, a_1^w) - r^*(x_1, a_1^\ell) - r(x_1, a_1^w) + r(x_1, a_1^\ell)\big|^2}$.

We have proved that with probability at least $1 - \delta$, the event $\mathcal{E} := \{\text{Eq. (43) holds for all } r \in \mathcal{N}_\epsilon(\mathcal{R}) \text{ simultaneously}\}$ occurs. We will extend the range to any $r \in \mathcal{R}$. By the definition of the $\epsilon$ cover $\mathcal{N}_\epsilon(\mathcal{R})$, there exists at least one $r^\dagger \in \mathcal{N}_\epsilon(\mathcal{R})$ such that $\|r^\dagger - r\|_\infty \leq \epsilon$. Therefore,

$$\big|\mathcal{L}_{N,\lambda}(r, \xi_r) - \mathcal{L}_{N,\lambda}(r^\dagger, \xi_{r^\dagger})\big|$$

$$\stackrel{(a)}{=} \Big|\frac{1}{N}\sum_{i=1}^N \big\{\log\sigma[r^\dagger(x_i, a_i^w) - r^\dagger(x_i, a_i^\ell) + \xi_{r^\dagger,i}] - \log\sigma[r(x_i, a_i^w) - r(x_i, a_i^\ell) + \xi_{r,i}]\big\}$$

$$+ \frac{\lambda}{N}(\|\xi_r\|_1 - \|\xi_{r^\dagger}\|_1)\Big|$$

$$\stackrel{(b)}{\leq} \frac{1}{N}\sum_{i=1}^N \Big[\big|[r^\dagger(x_i, a_i^w) - r^\dagger(x_i, a_i^\ell) + \xi_{r^\dagger,i}] - [r(x_i, a_i^w) - r(x_i, a_i^\ell) + \xi_{r,i}]\big| + \lambda(|\xi_{r,i}| - |\xi_{r^\dagger,i}|)\Big]$$

$$\leq \frac{1}{N}\sum_{i=1}^N \Big[\big|r^\dagger(x_i, a_i^w) - r(x_i, a_i^w)\big| + \big|r(x_i, a_i^\ell) - r^\dagger(x_i, a_i^\ell)\big| + \big|\xi_{r^\dagger,i} - \xi_{r,i}\big| + \lambda(|\xi_{r,i} - \xi_{r^\dagger,i}|)\Big]$$

$$\stackrel{(c)}{\leq} \frac{1}{N}\sum_{i=1}^N \Big[\big|r^\dagger(x_i, a_i^w) - r(x_i, a_i^w)\big| + \big|r(x_i, a_i^\ell) - r^\dagger(x_i, a_i^\ell)\big|$$

$$+ (\lambda+1)\big|r(x_i, a_i^\ell) - r(x_i, a_i^w) - [r^\dagger(x_i, a_i^\ell) - r^\dagger(x_i, a_i^w)]\big|\Big] \stackrel{(d)}{\leq} 6\epsilon, \tag{44}$$

where (a) uses the definition of $\mathcal{L}_{N,\lambda}$ given by Eq. (8), (b) uses triangle inequality and $\frac{d}{dv}[\log\sigma(v)] = \sigma(-v) \in [0,1]$ for any $v \in \mathcal{R}$, (c) uses the property that $\xi_{r,i}$ defined by Eq. (15) is a 1-Lipschitz continuous function of $r(x_i, a_i^\ell) - r(x_i, a_i^w)$ (since $\max(\cdot, 0)$ is 1-Lipschitz continuous), (d) uses $\|r^\dagger - r\|_\infty \leq \epsilon$ and $\lambda \leq 1$. Under the event $\mathcal{E}$, Eq. (43) holds with $r$ replaced by $r^+$, which along with Eq. (44) implies the following inequality.

$$\mathcal{L}_{N,\lambda}(r^*, \xi^*) - \mathcal{L}_{N,\lambda}(r, \xi_r)$$

$$\leq [\mathcal{L}_{N,\lambda}(r^\dagger, \xi_{r^\dagger}) - \mathcal{L}_{N,\lambda}(r, \xi_r)] + [\mathcal{L}_{N,\lambda}(r^*, \xi^*) - \mathcal{L}_{N,\lambda}(r^\dagger, \xi_{r^\dagger})]$$

$$\leq 6\epsilon + \frac{2}{N}\Big[\|\xi^*\|_1 + \log\Big(\frac{|\mathcal{N}_\epsilon(\mathcal{R})|}{\delta}\Big)\Big] - \frac{E_{r^\dagger}^2}{2(3+e^R)^2}$$

$$= 6\epsilon + \frac{2}{N}\Big[\|\xi^*\|_1 + \log\Big(\frac{|\mathcal{N}_\epsilon(\mathcal{R})|}{\delta}\Big)\Big] - \frac{E_{r^\dagger}^2 - E_r^2}{2(3+e^R)^2} - \frac{E_r^2}{2(3+e^R)^2}$$

$$\stackrel{(a)}{\leq} 6\epsilon + \frac{2}{N}\Big[\|\xi^*\|_1 + \log\Big(\frac{|\mathcal{N}_\epsilon(\mathcal{R})|}{\delta}\Big)\Big] + \frac{4R\epsilon}{(3+e^R)^2} - \frac{E_r^2}{2(3+e^R)^2}$$

$$\stackrel{(b)}{\leq} 7\epsilon + \frac{2}{N}\Big[\|\xi^*\|_1 + \log\Big(\frac{|\mathcal{N}_\epsilon(\mathcal{R})|}{\delta}\Big)\Big] - \frac{E_r^2}{2(3+e^R)^2}, \tag{45}$$

which proves Eq. (39). Here, (a) uses the following inequality and (b) uses $(3+e^R)^2 > 6e^R + e^{2R} > 6R + 2R = 8R$.

$$|E_{r^\dagger}^2 - E_r^2|$$

$$= \Big|\mathbb{E}_\mathcal{D}\big\{\big[r^*(x_1, a_1^w) - r^*(x_1, a_1^\ell) - r^\dagger(x_1, a_1^w) + r^\dagger(x_1, a_1^\ell)\big]^2\big\}$$

$$- \mathbb{E}_{\mathcal{D}}\big\{\big[r^*(x_1, a_1^w) - r^*(x_1, a_1^\ell) - r(x_1, a_1^w) + r(x_1, a_1^\ell)\big]^2\big\}\Big|$$

$$= \Big|\mathbb{E}_{\mathcal{D}}\big\{\big[r(x_1, a_1^w) - r(x_1, a_1^\ell) - r^\dagger(x_1, a_1^w) + r^\dagger(x_1, a_1^\ell)\big]$$

$$\big[2r^*(x_1, a_1^w) - 2r^*(x_1, a_1^\ell) - r^\dagger(x_1, a_1^w) + r^\dagger(x_1, a_1^\ell) - r(x_1, a_1^w) + r(x_1, a_1^\ell)\big]\big\}\Big|$$

$$\overset{(a)}{\leq} (2\epsilon)(4R) = 8R\epsilon,$$

where (a) uses Assumption 2 and $\|r^\dagger - r\|_\infty \leq \epsilon$. $\qquad\qquad\square$

**Lemma 9.** *Fixing any $\epsilon > 0$, $\delta \in (0,1)$, the online dataset $\{x_i, a_i^w, a_i^\ell, y_i\}_{i=1}^T$ generated from Algorithm 2 satisfies the following bound for all $t = 1, \ldots, T$ and $\pi \in \Pi_{\mathcal{R}} \overset{\text{def}}{=} \{\pi_r : r \in \mathcal{R}\}$ simultaneously with probability at least $1 - \delta$.*

$$\sum_{i=1}^t \log \frac{\sigma\big[r^\pi(x_i, a_i^w) - r^\pi(x_i, a_i^\ell) + y_i \xi_i^\pi\big]}{\sigma\big[r^*(x_i, a_i^w) - r^*(x_i, a_i^\ell) + y_i \xi_i^*\big]}$$

$$\leq 2\log\Big(\frac{T|\mathcal{N}_\epsilon(\mathcal{R})|}{\delta}\Big) + 4t\epsilon + \sum_{i=1}^t \Big\{\frac{1}{4}|\xi_i^*| + \sigma(R)|\xi_i^\pi|$$

$$- \frac{1}{2(3 + e^R)^2}\mathbb{E}_{x \sim \rho, a^{(1)} \sim \pi_i(\cdot|x), a^{(-1)} \sim \pi_{\text{ref}}(\cdot|x)}\big[f_\pi^2(x, a^{(1)}, a^{(-1)})\big]\Big\},$$

*where the function $f_\pi$ is defined below and $\mathcal{N}_\epsilon(\mathcal{R})$ is a finite $\epsilon$-cover of $\mathcal{R}$, that is, for any $r \in \mathcal{R}$, there exists $r^\dagger \in \mathcal{N}_\epsilon(\mathcal{R})$ satisfying $\|r^\dagger - r\|_\infty \leq \epsilon$.*

$$f_\pi(x, a^{(1)}, a^{(-1)}) \overset{\text{def}}{=} r^*(x, a^{(1)}) - r^*(x, a^{(-1)}) - r^\pi(x, a^{(1)}) + r^\pi(x, a^{(-1)}), \tag{46}$$

*Proof.* Define the following function.

$$q_{\pi, \xi_i}(y_i|x_i, a_i^{(1)}, a_i^{(-1)})$$

$$\overset{\text{def}}{=} \begin{cases} \sigma\Big(\beta \log \frac{\pi(a_i^{(1)}|x_i)}{\pi_{\text{ref}}(a_i^{(1)}|x_i)} - \beta \log \frac{\pi(a_i^{(-1)}|x_i)}{\pi_{\text{ref}}(a_i^{(-1)}|x_i)} + \omega(|a_i^{(1)}| - |a_i^{(-1)}|) + \xi_i\Big), & y_i = 1 \\ \sigma\Big(\beta \log \frac{\pi(a_i^{(-1)}|x_i)}{\pi_{\text{ref}}(a_i^{(-1)}|x_i)} - \beta \log \frac{\pi(a_i^{(1)}|x_i)}{\pi_{\text{ref}}(a_i^{(1)}|x_i)} + \omega(|a_i^{(-1)}| - |a_i^{(1)}|) - \xi_i\Big), & y_i = -1. \end{cases}$$

$$= \sigma\big[r^\pi(x_i, a_i^w) - r^\pi(x_i, a_i^\ell) + y_i \xi_i\big], \tag{47}$$

where the second $=$ uses Eq. (16) and merges the above two cases. The above $q_{\pi, \xi_i}(y_i|x_i, a_i^{(1)}, a_i^{(-1)})$ can be seen as a conditional probability of $y_i \in \{-1, 1\}$ since $q_{\pi, \xi_i}(1|x_i, a_i^{(1)}, a_i^{(-1)}) + q_{\pi, \xi_i}(-1|x_i, a_i^{(1)}, a_i^{(-1)}) = 1$.

Then define the following random variables for $i = 1, \ldots, T$.

$$W_i(\pi) = \frac{1}{2}\log \frac{\sigma\big[r^\pi(x_i, a_i^w) - r^\pi(x_i, a_i^\ell)\big]}{\sigma\big[r^*(x_i, a_i^w) - r^*(x_i, a_i^\ell) + y_i \xi_i^*\big]} = \frac{1}{2}\log \frac{q_{\pi, 0}(y_i|x_i, a_i^{(1)}, a_i^{(-1)})}{p_{r^*, \xi_i^*}(y_i|x_i, a_i^{(1)}, a_i^{(-1)})}, \tag{48}$$

where $p_{r, \xi_i}(y_i|x_i, a_i^{(1)}, a_i^{(-1)})$ is defined by Eq. (41).

For any $r \in \mathcal{R}$, there exists $r^\dagger \in \mathcal{N}_\epsilon(\mathcal{R})$ satisfying $\|r^\dagger - r\|_\infty \leq \epsilon$, and thus we can temporarily denote $r_u = ur^{\pi_{r^\dagger}} + (1 - u)r^\pi$ $(u \in [0,1])$. Then we obtain that

$$\Big|\frac{d}{du}\log \sigma\big[r_u(x_i, a_i^w) - r_u(x_i, a_i^\ell)\big]\Big|$$

$$= \sigma\big[r_u(x_i, a_i^\ell) - r_u(x_i, a_i^w)\big]\big|r^{\pi_{r^\dagger}}(x_i, a_i^w) - r^{\pi_{r^\dagger}}(x_i, a_i^\ell) - r^\pi(x_i, a_i^w) + r^\pi(x_i, a_i^\ell)\big|$$

$$\overset{(a)}{\leq} \big|r^\dagger(x_i, a_i^w) - r^\dagger(x_i, a_i^\ell) - r^\pi(x_i, a_i^w) + r^\pi(x_i, a_i^\ell)\big|$$

$$\leq \big|r^\dagger(x_i, a_i^w) - r^\pi(x_i, a_i^w)\big| + \big|r^\pi(x_i, a_i^\ell) - r^\dagger(x_i, a_i^\ell)\big| \leq 2\epsilon, \tag{49}$$

where (a) uses Eq. (29) and $\sigma(x) \in (0,1)$ for any $x \in \mathbb{R}$. Therefore,

$$
\begin{aligned}
&|W_i(\pi_{r^\dagger}) - W_i(\pi)| \\
&\overset{(a)}{=} \frac{1}{2}\Big[ \log q_{\pi_{r^\dagger},0}(y_i|x_i, a_i^{(1)}, a_i^{(-1)}) - \log q_{\pi,0}(y_i|x_i, a_i^{(1)}, a_i^{(-1)}) \Big] \\
&\overset{(b)}{=} \frac{1}{2}\Big| \log \sigma\big[r^{\pi_{r^\dagger}}(x_i, a_i^w) - r^{\pi_{r^\dagger}}(x_i, a_i^\ell)\big] - \log \sigma\big[r^\pi(x_i, a_i^w) - r^\pi(x_i, a_i^\ell)\big] \Big| \\
&\overset{(c)}{=} \frac{1}{2}\Big| \log \sigma\big[r_1(x_i, a_i^w) - r_1(x_i, a_i^\ell)\big] - \log \sigma\big[r_0(x_i, a_i^w) - r_0(x_i, a_i^\ell)\big] \Big| \overset{(d)}{\leq} \epsilon,
\end{aligned}
\tag{50}
$$

where (a) and (b) use Eq. (48), (c) uses the above notation that $r_u = u r^{\pi_{r^\dagger}} + (1-u) r^\pi$ ($u \in [0,1]$), and (d) uses Eq. (49). Then based on Algorithm 2 and Assumption 1, given $(x_i, a_i^{(1)}, a_i^{(-1)})$, the label $y_i$ is generated with probability distribution $p_{r^*, \xi_i}(y_i|x_i, a_i^{(1)}, a_i^{(-1)})$ defined by Eq. (41). Therefore, given any $\delta \in (0,1)$ and $\epsilon > 0$, by Lemma 7 with $\lambda' = 1$, the following inequality holds for $t = 1, \ldots, T$ and finitely many $\pi' \in \mathcal{N}_\epsilon(\mathcal{R})$ simultaneously with probability at least $1 - \delta$.

$$
\sum_{i=1}^t W_i(\pi') \leq \log\Big(\frac{T|\mathcal{N}_\epsilon(\mathcal{R})|}{\delta}\Big) + \sum_{i=1}^t \log \mathbb{E}_{\mu_i}[e^{W_i(\pi')}].
$$

where $\mu_i$ denotes the distribution of the $i$-th online data sample $(x_i, a_i^{(-1)}, a_i^{(1)}, y_i)$ generated by Algorithm 2. We further upper bound the above inequality as follows.

$$
\begin{aligned}
&\sum_{i=1}^t W_i(\pi') - \log\Big(\frac{T|\mathcal{N}_\epsilon(\mathcal{R})|}{\delta}\Big) \\
&\leq \sum_{i=1}^t \log \mathbb{E}_{\mu_i}[e^{W_i(\pi')}] \\
&\overset{(48)}{=} \sum_{i=1}^t \log \mathbb{E}_{\mu_i}\Bigg\{ \mathbb{E}_{y_i \sim p_{r^*, \xi_i^*}(\cdot|x_i, a_i^{(1)}, a_i^{(-1)})}\Bigg[ \sqrt{\frac{q_{\pi',0}(y_i|x_i, a_i^{(1)}, a_i^{(-1)})}{p_{r^*, \xi_i^*}(y_i|x_i, a_i^{(1)}, a_i^{(-1)})}} \Bigg| x_i, a_i^{(1)}, a_i^{(-1)} \Bigg] \Bigg\} \\
&\overset{(a)}{\leq} \sum_{i=1}^t \mathbb{E}_{\mu_i}\Bigg[ \sum_{y \in \{-1,1\}} \sqrt{q_{\pi',0}(y|x_i, a_i^{(1)}, a_i^{(-1)}) p_{r^*, \xi_i^*}(y|x_i, a_i^{(1)}, a_i^{(-1)})} - 1 \Bigg] \\
&= -\frac{1}{2}\sum_{i=1}^t \mathbb{E}_{\mu_i}\Bigg[ \sum_{y \in \{-1,1\}} \Big| \sqrt{q_{\pi',0}(y|x_i, a_i^{(1)}, a_i^{(-1)})} - \sqrt{p_{r^*, \xi_i^*}(y|x_i, a_i^{(1)}, a_i^{(-1)})} \Big|^2 \Bigg] \\
&\overset{(b)}{\leq} -\frac{1}{8}\sum_{i=1}^t \mathbb{E}_{\mu_i}\Bigg[ \sum_{y \in \{-1,1\}} \big| q_{\pi',0}(y|x_i, a_i^{(1)}, a_i^{(-1)}) - p_{r^*, \xi_i^*}(y|x_i, a_i^{(1)}, a_i^{(-1)}) \big|^2 \Bigg] \\
&\overset{(c)}{=} -\frac{1}{4}\sum_{i=1}^t \mathbb{E}_{\mu_i}\Big\{ \sigma[r^{\pi'}(x_i, a_i^w) - r^{\pi'}(x_i, a_i^\ell)] - \sigma[r^*(x_i, a_i^w) - r^*(x_i, a_i^\ell) + y_i \xi_i^*] \Big\}^2 \\
&\overset{(d)}{\leq} -\frac{1}{4}\sum_{i=1}^t \Big\{ \Big[ \mathbb{E}_{\mu_i}[\sigma[r^{\pi_{r^*}}(x_i, a_i^w) - r^{\pi_{r^*}}(x_i, a_i^\ell)] - \sigma[r^{\pi'}(x_i, a_i^w) - r^{\pi'}(x_i, a_i^\ell)]]^2 \Big] - \frac{1}{2}|\xi_i^*| \Big\} \\
&\overset{(e)}{\leq} \frac{1}{8}\sum_{i=1}^t \Big\{ |\xi_i^*| - \frac{2}{(3+e^R)^2} \mathbb{E}_{\mu_i}\Big[ |r^{\pi_{r^*}}(x_i, a_i^w) - r^{\pi_{r^*}}(x_i, a_i^\ell) - r^{\pi'}(x_i, a_i^w) + r^{\pi'}(x_i, a_i^\ell)|^2 \Big] \Big\},
\end{aligned}
\tag{51}
$$

where (a) uses $\log v \leq v - 1$ for any $v > 0$, (b) uses Lemma 12.2 of (Harsha, 2011), (c) uses Eqs. (41) and (47), (d) uses Eq. (29) and Lemma 5, and (e) uses Assumption 2 and Lemma 1. Combining Eqs. (50) and (51), we obtain the following inequality which holds for all $t = 1, \ldots, T$ and $\pi \in \Pi$ simultaneously with probability at least $1 - \delta$.

$$
\sum_{i=1}^t W_i(\pi)
$$

$$\leq \sum_{i=1}^{t} [W_i(\pi) - W_i(\pi_{r^\dagger})] + W_i(\pi_{r^\dagger})$$

$$\overset{(a)}{\leq} \frac{1}{8} \sum_{i=1}^{t} \left\{ |\xi_i^*| - \frac{2}{(3+e^R)^2} \mathbb{E}_{\mu_i} \left[ \left[ r^{\pi_{r^*}}(x_i, a_i^w) - r^{\pi_{r^*}}(x_i, a_i^\ell) - r^{\pi_{r^\dagger}}(x_i, a_i^w) + r^{\pi_{r^\dagger}}(x_i, a_i^\ell) \right]^2 \right] \right\}$$

$$+ \log\left( \frac{T|\mathcal{N}_\epsilon(\mathcal{R})|}{\delta} \right) + t\epsilon$$

$$\overset{(b)}{\leq} \frac{1}{8} \sum_{i=1}^{t} \left\{ |\xi_i^*| - \frac{2}{(3+e^R)^2} \mathbb{E}_{\mu_i} \left[ \left[ r^{\pi_{r^*}}(x_i, a_i^w) - r^{\pi_{r^*}}(x_i, a_i^\ell) - r^{\pi}(x_i, a_i^w) + r^{\pi}(x_i, a_i^\ell) \right]^2 \right] \right\}$$

$$+ \log\left( \frac{T|\mathcal{N}_\epsilon(\mathcal{R})|}{\delta} \right) + 2t\epsilon, \tag{52}$$

where (a) uses Eq. (51) (with $\pi'$ replaced by $\pi_{r^\dagger}$) and Eq. (50), (b) uses the following inequality and $(3+e^R)^2 > 6e^R + e^{2R} > 6R + 2R = 8R$.

$$\left[ r^{\pi_{r^*}}(x_i, a_i^w) - r^{\pi_{r^*}}(x_i, a_i^\ell) - r^{\pi}(x_i, a_i^w) + r^{\pi}(x_i, a_i^\ell) \right]^2$$

$$- \left[ r^{\pi_{r^*}}(x_i, a_i^w) - r^{\pi_{r^*}}(x_i, a_i^\ell) - r^{\pi_{r^\dagger}}(x_i, a_i^w) + r^{\pi_{r^\dagger}}(x_i, a_i^\ell) \right]^2$$

$$= \left[ r^{\pi_{r^\dagger}}(x_i, a_i^w) - r^{\pi_{r^\dagger}}(x_i, a_i^\ell) - r^{\pi}(x_i, a_i^w) + r^{\pi}(x_i, a_i^\ell) \right]$$

$$\left[ 2r^{\pi_{r^*}}(x_i, a_i^w) - 2r^{\pi_{r^*}}(x_i, a_i^\ell) - r^{\pi}(x_i, a_i^w) + r^{\pi}(x_i, a_i^\ell) - r^{\pi_{r^\dagger}}(x_i, a_i^w) + r^{\pi_{r^\dagger}}(x_i, a_i^\ell) \right]$$

$$\overset{(a)}{=} \left[ r^{\dagger}(x_i, a_i^w) - r^{\dagger}(x_i, a_i^\ell) - r^{\pi}(x_i, a_i^w) + r^{\pi}(x_i, a_i^\ell) \right]$$

$$\left[ 2r^{\pi_{r^*}}(x_i, a_i^w) - 2r^{\pi_{r^*}}(x_i, a_i^\ell) - r^{\pi}(x_i, a_i^w) + r^{\pi}(x_i, a_i^\ell) - r^{\pi_{r^\dagger}}(x_i, a_i^w) + r^{\pi_{r^\dagger}}(x_i, a_i^\ell) \right]$$

$$\overset{(b)}{\leq} (2\epsilon)(4R) = 8R\epsilon,$$

where (a) uses Eq. (29), and (b) uses $\|r^\dagger - r\|_\infty \leq \epsilon$ and Lemma 3.

Finally, we conclude the proof as follows.

$$\sum_{i=1}^{t} \log \frac{\sigma\left[ r^{\pi}(x_i, a_i^w) - r^{\pi}(x_i, a_i^\ell) + y_i \xi_i^\pi \right]}{\sigma\left[ r^*(x_i, a_i^w) - r^*(x_i, a_i^\ell) + y_i \xi_i^* \right]}$$

$$\overset{(a)}{\leq} \sum_{i=1}^{t} \left[ \log \frac{\sigma\left[ r^{\pi}(x_i, a_i^w) - r^{\pi}(x_i, a_i^\ell) \right]}{\sigma\left[ r^*(x_i, a_i^w) - r^*(x_i, a_i^\ell) + y_i \xi_i^* \right]} + \sigma(R)|\xi_i^\pi| \right]$$

$$\overset{(b)}{=} \sum_{i=1}^{t} \left[ 2W_i(\pi) + \sigma(R)|\xi_i^\pi| \right]$$

$$\overset{(c)}{\leq} 2\log\left( \frac{T|\mathcal{N}_\epsilon(\mathcal{R})|}{\delta} \right) + 4t\epsilon + \sum_{i=1}^{t} \left\{ \frac{1}{4}|\xi_i^*| + \sigma(R)|\xi_i^\pi| \right.$$

$$\left. - \frac{1}{2(3+e^R)^2} \mathbb{E}_{\mu_i} \left[ \left[ r^{\pi_{r^*}}(x_i, a_i^w) - r^{\pi_{r^*}}(x_i, a_i^\ell) - r^{\pi}(x_i, a_i^w) + r^{\pi}(x_i, a_i^\ell) \right]^2 \right] \right\}$$

$$\overset{(d)}{=} 2\log\left( \frac{T|\mathcal{N}_\epsilon(\mathcal{R})|}{\delta} \right) + 4t\epsilon + \sum_{i=1}^{t} \left\{ \frac{1}{4}|\xi_i^*| + \sigma(R)|\xi_i^\pi| \right.$$

$$\left. - \frac{1}{2(3+e^R)^2} \mathbb{E}_{\mu_i} \left[ \left[ r^*(x_i, a_i^{(1)}) - r^*(x_i, a_i^{(-1)}) - r^{\pi}(x_i, a_i^{(1)}) + r^{\pi}(x_i, a_i^{(-1)}) \right]^2 \right] \right\}$$

$$\overset{(e)}{=} 2\log\left( \frac{T|\mathcal{N}_\epsilon(\mathcal{R})|}{\delta} \right) + 4t\epsilon + \sum_{i=1}^{t} \left\{ \frac{1}{4}|\xi_i^*| + \sigma(R)|\xi_i^\pi| \right.$$

$$\left. - \frac{1}{2(3+e^R)^2} \mathbb{E}_{x\sim\rho, a^{(1)}\sim\pi_i(\cdot|x), a^{(-1)}\sim\pi_{\text{ref}}(\cdot|x)} \left[ f_\pi^2(x, a^{(1)}, a^{(-1)}) \right] \right\},$$

where (a) uses Eq. (33) from Lemma 4, (b) uses $W_i(\pi)$ defined by Eq. (48), (c) uses Eq. (52), (d) uses Eq. (29) and $\{a_i^w, a_i^\ell\} = \{a_i^{(1)}, a_i^{(-1)}\}$ (based on Assumption 1), and (e) uses Eq. (46). $\qquad\square$

**Lemma 10** (Azuma-Hoeffding Inequality (Xie et al., 2024)). *The random variables $\{X_t\}_{t=1}^T$ satisfy $|X_t| \leq C$ almost surely. Then with probability at least $1 - \delta$, we have*

$$\Big| \sum_{t=1}^T [X_t - \mathbb{E}(X_t | X_1, \ldots, X_{t-1})] \Big| \leq C\sqrt{8T \log(2/\delta)}. \tag{53}$$

**Lemma 11.** *Fixing any $\epsilon > 0$, $\delta \in (0, 1)$, the online dataset $\{x_i, a_i^{(1)}, a_i^{(-1)}, y_i\}_{i=1}^T$ generated from Algorithm 2 satisfies the following inequality for all $t = 1, \ldots, T$ and $\pi \in \Pi_{\mathcal{R}} \overset{\text{def}}{=} \{\pi_r : r \in \mathcal{R}\}$ simultaneously with probability at least $1 - \delta$.*

$$\left| \Big[ \sum_{i=1}^t \log \frac{\pi(a_i^{(-1)}|x_i)}{\pi_{r^*}(a_i^{(-1)}|x_i)} \Big] - t\mathbb{E}_{x\sim\rho, a\sim\pi_{\text{ref}}(\cdot|x)}\Big[ \log \frac{\pi(a|x)}{\pi_{r^*}(a|x)} \Big] \right| \leq \frac{4R}{\beta}\sqrt{2t \log\Big[ \frac{2T\mathcal{N}_\epsilon(\mathcal{R})}{\delta} \Big]} + \frac{4t\epsilon}{\beta}.$$

*Proof.* For any $r \in \mathcal{R}$, denote $X_i(r) = \log \frac{\pi_r(a_i^{(-1)}|x_i)}{\pi_{r^*}(a_i^{(-1)}|x_i)}$ which satisfies $|X_i(r)| \leq \frac{2R}{\beta}$ based on Lemma 6 and Assumption 2.

Then by applying Lemma 10 to $X_i(r)$ with union bound, we obtain the following inequality which holds for all $t = 0, 1, \ldots, T - 1$ and $r' \in \mathcal{N}_\epsilon(\mathcal{R})$ simultaneously with probability at least $1 - \delta$.

$$\left| \sum_{i=1}^t [X_i(r') - \mathbb{E}_{\mu_i} X_i(r')] \right| \leq \frac{2R}{\beta}\sqrt{8t \log\Big[ \frac{2T\mathcal{N}_\epsilon(\mathcal{R})}{\delta} \Big]}. \tag{54}$$

where $\mu_i$ denotes the distribution of the $i$-th online data sample $(x_i, a_i^{(-1)}, a_i^{(1)}, y_i)$ generated by Algorithm 2.

For any $r \in \mathcal{R}$, there exists $r^\dagger \in \mathcal{N}_\epsilon(\mathcal{R})$ satisfying $\|r^\dagger - r\|_\infty \leq \epsilon$, so Lemma 6 implies that

$$|X_i(r^\dagger) - X_i(r)| = \left| \log \frac{\pi_{r^\dagger}(a_i^{(-1)}|x_i)}{\pi_r(a_i^{(-1)}|x_i)} \right| \leq \frac{2\epsilon}{\beta}.$$

Therefore, if the above high probability event $\mathcal{E} := \{$Eq. (54) holds for all $r' \in \mathcal{N}_\epsilon(\mathcal{R})\}$ occurs, then the following inequality holds for any $r \in \mathcal{R}$.

$$\left| \sum_{i=1}^t [X_i(r) - \mathbb{E}_{\mu_i} X_i(r)] \right| \leq \frac{2R}{\beta}\sqrt{8t \log\Big[ \frac{2T\mathcal{N}_\epsilon(\mathcal{R})}{\delta} \Big]} + \frac{4t\epsilon}{\beta}. \tag{55}$$

For any $\pi \in \Pi_{\mathcal{R}} \overset{\text{def}}{=} \{\pi_r : r \in \mathcal{R}\}$, there exists $r \in \mathcal{R}$ satisfying $\pi = \pi_r$. Then we have

$$X_i(r) = \log \frac{\pi(a_i^{(-1)}|x_i)}{\pi_{r^*}(a_i^{(-1)}|x_i)}.$$

and thus

$$\mathbb{E}_{\mu_i} X_i(r) = \mathbb{E}_{x_i\sim\rho, a_i^{(-1)}\sim\pi_{\text{ref}}(\cdot|x)}\left[ \log \frac{\pi(a_i^{(-1)}|x_i)}{\pi_{r^*}(a_i^{(-1)}|x_i)} \right] = \mathbb{E}_{x\sim\rho, a\sim\pi_{\text{ref}}(\cdot|x)}\left[ \log \frac{\pi(a|x)}{\pi_{r^*}(a|x)} \right].$$

Substituting the above two equalities into Eq. (55) concludes the proof. $\qquad\square$

**Lemma 12.** *Suppose that the offline dataset $\{x_i, a_i^w, a_i^\ell, y_i\}_{i=1}^N$ is generated from Assumption 1, and select the baseline policy $\pi_{\text{base}}$ to be the distribution of $a_i^w$ given $x_i$. Then fixing any $\epsilon > 0$, $\delta \in (0, 1)$, the following inequality holds for all $\pi \in \Pi_{\mathcal{R}} \overset{\text{def}}{=} \{\pi_r : r \in \mathcal{R}\}$ simultaneously with probability at least $1 - \delta$.*

$$\left| \Big[ \sum_{i=1}^N \log \frac{\pi(a_i^w|x_i)}{\pi_{r^*}(a_i^w|x_i)} \Big] - N\mathbb{E}_{x\sim\rho, a\sim\pi_{\text{base}}(\cdot|x)}\Big[ \log \frac{\pi(a|x)}{\pi_{r^*}(a|x)} \Big] \right| \leq \frac{4R}{\beta}\sqrt{2N \log\Big[ \frac{2\mathcal{N}_\epsilon(\mathcal{R})}{\delta} \Big]} + \frac{4N\epsilon}{\beta}.$$

*Proof.* The proof logic is the same as that of Lemma 11. The major difference is that the inequality here only has to hold for any $\pi \in \Pi_{\mathcal{R}}$ while Lemma 11 requires to hold also for $t = 1, \ldots, T$. As a result, when applying Lemma 10 with union bound, $\frac{2T\mathcal{N}_\epsilon(\mathcal{R})}{\delta}$ in the proof of Lemma 11 is replaced with $\frac{2\mathcal{N}_\epsilon(\mathcal{R})}{\delta}$. $\qquad\square$

**Lemma 13.** *Define the following quantity.*

$$I_t \overset{\text{def}}{=} \frac{\left[\mathbb{E}_{x\sim\rho, a^{(1)}\sim\pi_{t+1}(\cdot|x), a^{(-1)}\sim\pi_{\text{ref}}(\cdot|x)} f_{\pi_{t+1}}(x, a^{(1)}, a^{(-1)})\right]^2}{R^2 + \sum_{i=1}^t \mathbb{E}_{x\sim\rho, a^{(1)}\sim\pi_i(\cdot|x), a^{(-1)}\sim\pi_{\text{ref}}(\cdot|x)}[f_{\pi_{t+1}}^2(x, a^{(1)}, a^{(-1)})]}, \tag{56}$$

*where the function $f_\pi$ is defined by Eq. (46). Then we have*

$$\sum_{t=1}^T I_t \leq 12 G_{\text{on}} \log(T + 2), \tag{57}$$

*where $G_{\text{on}}$ is defined by Eq. (26).*

*Proof.* Applying Assumption 2 and Lemma 3 to the function $f_\pi$ defined by Eq. (46), we have

$$f_\pi(x, a^{(1)}, a^{(-1)}) = r^*(x, a^{(1)}) - r^*(x, a^{(-1)}) - r^\pi(x, a^{(1)}) + r^\pi(x, a^{(-1)}) \in [-2R, 2R]. \tag{58}$$

Denote $\nu^* \in \text{argmin}_{\nu\in\Pi_{\mathcal{R}}} \sup_{x\in\mathcal{X}, a\in\mathcal{A}, \pi\in\Pi_{\mathcal{R}}} \frac{\pi(a|x)}{\nu(a|x)}$ as the policy used in the coverability coefficient (26). Then we have

$$\pi(a^{(1)}|x) \leq G_{\text{on}}\nu^*(a^{(1)}|x), \quad \forall x \in \mathcal{X}, a^{(1)} \in \mathcal{A}, \pi \in \Pi_{\mathcal{R}}. \tag{59}$$

Then for each $(x, a^{(1)}) \in \mathcal{X} \times \mathcal{A}$, define the following quantity ($\min \emptyset = +\infty$ by default)

$$\tau(x, a^{(1)}) = \min\left\{t \geq 1 \,\middle|\, \sum_{i=1}^t \pi_{i+1}(a^{(1)}|x) \geq G_{\text{on}}\nu^*(a^{(1)}|x)\right\}. \tag{60}$$

Hence,

$$\sum_{t=1}^T \pi_{t+1}(a^{(1)}|x)\mathbb{I}\{t \leq \tau(x, a^{(1)}) - 1\} < G_{\text{on}}\nu^*(a^{(1)}|x), \tag{61}$$

$$\sum_{i=1}^t \pi_i(a^{(1)}|x) \geq G_{\text{on}}\nu^*(a^{(1)}|x), \qquad \forall t \geq \tau(x, a^{(1)}) + 1. \tag{62}$$

Then we conclude the proof as follows.

$$\sum_{t=1}^T I_t$$

$$= \sum_{t=1}^T \frac{\left[\mathbb{E}_{x\sim\rho, a^{(1)}\sim\pi_{t+1}(\cdot|x), a^{(-1)}\sim\pi_{\text{ref}}(\cdot|x)} f_{\pi_{t+1}}(x, a^{(1)}, a^{(-1)})\mathbb{I}\{t \leq \tau(x, a^{(1)})\}\right]^2}{R^2 + \sum_{i=1}^t \mathbb{E}_{x\sim\rho, a^{(1)}\sim\pi_i(\cdot|x), a^{(-1)}\sim\pi_{\text{ref}}(\cdot|x)}[f_{\pi_{t+1}}^2(x, a^{(1)}, a^{(-1)})]}$$

$$+ \sum_{t=1}^T \frac{\left[\mathbb{E}_{x\sim\rho, a^{(1)}\sim\pi_{t+1}(\cdot|x), a^{(-1)}\sim\pi_{\text{ref}}(\cdot|x)} f_{\pi_{t+1}}(x, a^{(1)}, a^{(-1)})\mathbb{I}\{t \geq \tau(x, a^{(1)}) + 1\}\right]^2}{R^2 + \sum_{i=1}^t \mathbb{E}_{x\sim\rho, a^{(1)}\sim\pi_i(\cdot|x), a^{(-1)}\sim\pi_{\text{ref}}(\cdot|x)}[f_{\pi_{t+1}}^2(x, a^{(1)}, a^{(-1)})]}$$

$$\overset{(a)}{\leq} \frac{1}{R^2} \sum_{t=1}^T (2R\mathbb{E}_{x\sim\rho, a^{(1)}\sim\pi_{t+1}(\cdot|x)}\mathbb{I}\{t \leq \tau(x, a^{(1)})\})^2$$

$$+ \sum_{t=1}^T \frac{\left[\mathbb{E}_{x\sim\rho, a^{(1)}\sim\overline{\pi}_t(\cdot|x), a^{(-1)}\sim\pi_{\text{ref}}(\cdot|x)} f_{\pi_{t+1}}(x, a^{(1)}, a^{(-1)}) \cdot \frac{\pi_{t+1}(a^{(1)}|x)}{\overline{\pi}_t(a^{(1)}|x)}\mathbb{I}\{t \geq \tau(x, a^{(1)}) + 1\}\right]^2}{t\mathbb{E}_{x\sim\rho, a^{(1)}\sim\overline{\pi}_t(\cdot|x), a^{(-1)}\sim\pi_{\text{ref}}(\cdot|x)}[f_{\pi_{t+1}}^2(x, a^{(1)}, a^{(-1)})]}$$

$$\overset{(b)}{\leq} 4 \sum_{t=1}^{T} \mathbb{E}_{x\sim\rho, a^{(1)}\sim\pi_{t+1}(\cdot|x)} \mathbb{I}\{t \leq \tau(x, a^{(1)})\}$$

$$+ \sum_{t=1}^{T} \frac{1}{t} \mathbb{E}_{x\sim\rho, a^{(1)}\sim\overline{\pi}_t(\cdot|x)} \left[\frac{\pi_{t+1}(a^{(1)}|x)}{\overline{\pi}_t(a^{(1)}|x)}\right]^2 \mathbb{I}\{t \geq \tau(x, a^{(1)}) + 1\}$$

$$= 4 \sum_{x,a^{(1)}} \rho(x) \left[\sum_{t=1}^{T}[\pi_{t+1}(a^{(1)}|x)\mathbb{I}\{t \leq \tau(x, a^{(1)}) - 1\}] + \sum_{t=1}^{T}[\pi_{t+1}(a^{(1)}|x)\mathbb{I}\{t = \tau(x, a^{(1)})\}]\right]$$

$$+ 2 \sum_{x,a^{(1)}} \rho(x) \sum_{t=1}^{T} \frac{\pi_{t+1}(a^{(1)}|x)}{t\overline{\pi}_t(a^{(1)}|x) + t\overline{\pi}_t(a^{(1)}|x)}[\pi_{t+1}(a^{(1)}|x)\mathbb{I}\{t \geq \tau(x, a^{(1)}) + 1\}]$$

$$\overset{(c)}{\leq} 4 \sum_{x,a^{(1)}} \rho(x)[G_{\mathrm{on}}\nu^*(a^{(1)}|x) + G_{\mathrm{on}}\nu^*(a^{(1)}|x)]$$

$$+ 2 \sum_{x,a^{(1)}} \rho(x) \sum_{t=1}^{T} \frac{\pi_{t+1}(a^{(1)}|x)}{t\overline{\pi}_t(a^{(1)}|x) + G_{\mathrm{on}}\nu^*(a^{(1)}|x)}[\pi_{t+1}(a^{(1)}|x)\mathbb{I}\{t \geq \tau(x, a^{(1)}) + 1\}]$$

$$\overset{(d)}{\leq} 8G_{\mathrm{on}} \sum_{x,a^{(1)}} \rho(x)\nu^*(a^{(1)}|x)$$

$$+ 4 \sum_{x,a^{(1)}} \rho(x) \sum_{t=1}^{T} \log\left[\frac{(t+1)\overline{\pi}_{t+1}(a^{(1)}|x) + G_{\mathrm{on}}\nu^*(a^{(1)}|x)}{t\overline{\pi}_t(a^{(1)}|x) + G_{\mathrm{on}}\nu^*(a^{(1)}|x)}\right][G_{\mathrm{on}}\nu^*(a^{(1)}|x)]$$

$$= 8G_{\mathrm{on}} + 4G_{\mathrm{on}} \sum_{x,a^{(1)}} \rho(x)\nu^*(a^{(1)}|x) \log\left[\frac{(T+1)\overline{\pi}_{T+1}(a^{(1)}|x) + G_{\mathrm{on}}\nu^*(a^{(1)}|x)}{\overline{\pi}_1(a^{(1)}|x) + G_{\mathrm{on}}\nu^*(a^{(1)}|x)}\right]$$

$$\overset{(e)}{\leq} 8G_{\mathrm{on}} + 4G_{\mathrm{on}} \sum_{x,a^{(1)}} \rho(x)\nu^*(a^{(1)}|x) \log\left[\frac{(T+1)G_{\mathrm{on}}\nu^*(a^{(1)}|x) + G_{\mathrm{on}}\nu^*(a^{(1)}|x)}{G_{\mathrm{on}}\nu^*(a^{(1)}|x)}\right]$$

$$\leq 12G_{\mathrm{on}} \log(T+2),$$

where (a) denotes $\overline{\pi}_t = \frac{1}{t}\sum_{i=1}^{t}\pi_i$ and uses Eq. (58) and $(\mathbb{E}X)^2 \leq \mathbb{E}(X^2)$ for any random variable $X \in \mathbb{R}$, (b) uses Cauchy-Schwartz inequality, (c) uses Eqs. (59), (61) and Eq. (62), (d) uses Eq. (59) and the inequality that $u \leq 2\log(1+u)$ for $u = \frac{\pi_{t+1}(a^{(1)}|x)}{t\overline{\pi}_t(a^{(1)}|x)+G_{\mathrm{on}}\nu^*(a^{(1)}|x)} \in [0,1]$ ($u \in [0,1]$ due to Eq. (59)), (e) uses Eq. (59). $\qquad \square$

## C. Proof of Proposition 1

$(\pi, r, \xi)$ is the solution to the offline RLHF-COV objective (12) means the following two conditions hold

$$\pi \in \arg\max_{\pi'\in\Pi} \mathcal{L}_{N,\lambda}(r, \xi) + \eta V_{\beta,\omega}(\pi', r),$$
$$(r, \xi) \in \arg\min_{r'\in\mathcal{R}, \xi'\in\mathbb{R}^N} \max_{\pi'\in\Pi} \mathcal{L}_{N,\lambda}(r', \xi') + \eta V_{\beta,\omega}(\pi', r').$$

Based on the notation that $\pi_r \overset{\mathrm{def}}{=} \arg\max_{\pi'\in\Pi} V_{\beta,\omega}(\pi', r)$, the above two conditions are equivalent to

$$\pi = \pi_r, \quad (r, \xi) \in \arg\min_{r'\in\mathcal{R}, \xi'\in\mathbb{R}^N} \mathcal{L}_{N,\lambda}(r', \xi') + \eta V_{\beta,\omega}(\pi_{r'}, r')$$

Furthermore, based on the notation that $\xi_r \overset{\mathrm{def}}{=} \arg\min_{\xi\in\mathbb{R}^N} \mathcal{L}_{N,\lambda}(r, \xi)$, the above two conditions are equivalent to

$$\pi = \pi_r, \quad \xi = \xi_r, \quad r = \arg\min_{r'\in\mathcal{R}} \mathcal{L}_{N,\lambda}(r', \xi_{r'}) + \eta V_{\beta,\omega}(\pi_{r'}, r'). \tag{63}$$

This prove the first part of the theorem.

Next, we will obtain the analytical solutions of $\pi_r$ and $\xi_{r,i}$. We rewrite the function (11) as follows.

$$V_{\beta,\omega}(\pi, r)$$

$$=\mathbb{E}_{x\sim\rho,a\sim\pi(\cdot|x),a'\sim\pi_{\text{base}}(\cdot|x)}\big[r(x,a)+\omega|a|-r(x,a')-\omega|a'|\big]-\beta\mathbb{E}_{x\sim\rho}\text{KL}\big[\pi(\cdot|x)\big\|\pi_{\text{ref}}(\cdot|x)\big]$$

$$=\mathbb{E}_{x\sim\rho,a\sim\pi(\cdot|x)}\Big[r(x,a)+\omega|a|-\beta\log\frac{\pi(a|x)}{\pi_{\text{ref}}(a|x)}\Big]-\mathbb{E}_{x\sim\rho,a'\sim\pi_{\text{base}}(\cdot|x)}\big[r(x,a')+\omega|a'|\big]$$

$$=-\beta\mathbb{E}_{x\sim\rho,a\sim\pi(\cdot|x)}\Big[\log\frac{\pi(a|x)/Z_r(x)}{\pi_{\text{ref}}(a|x)\exp\big[[r(x,a)+\omega|a|]/\beta\big]/Z_r(x)}\Big]$$

$$\quad-\mathbb{E}_{x\sim\rho,a'\sim\pi_{\text{base}}(\cdot|x)}\big[r(x,a')+\omega|a'|\big]$$

$$=C-\beta\mathbb{E}_{x\sim\rho}\text{KL}\Big[\pi(\cdot|x)\Big\|\pi_{\text{ref}}(\cdot|x)\exp\big[[r(x,\cdot)+\omega|\cdot|]/\beta\big]/Z_r(x)\Big],$$

where $Z_r(x)\stackrel{\text{def}}{=}\sum_{a'\in\mathcal{A}}\pi_{\text{ref}}(a'|x)\exp\big[\frac{r(x,a')-\omega|a'|}{\beta}\big]$ and the constant $C=\beta\mathbb{E}_{x\sim\rho}\log Z_r(x)-\mathbb{E}_{x\sim\rho,a'\sim\pi_{\text{base}}(\cdot|x)}\big[r(x,a')+\omega|a'|\big]$ is independent of $\pi$. Therefore, $\pi_r\stackrel{\text{def}}{=}\arg\max_{\pi'\in\Pi}V_{\beta,\omega}(\pi',r)$ should minimize the above KL term, which gives the analytical solution (14).

Note that the log-likelihood function (8) can be rewritten as follows.

$$\mathcal{L}_{N,\lambda}(r,\xi)\stackrel{\text{def}}{=}\frac{1}{N}\sum_{i=1}^{N}f_i(\xi_i),$$

where $f_i(v):=\lambda|v|-\log\sigma[r(x_i,a_i^w)-r(x_i,a_i^\ell)+y_iv]$. Hence, $\xi_r\in\arg\min_\xi\mathcal{L}_{N,\lambda}(r,\xi)$ is equivalent to the following condition:

$$\xi_{r,i}\in\arg\min_{v\in\mathbb{R}}f_i(v);\,i=1,2,\ldots,N.$$

As $f_i$ is a convex function for $\lambda>0$, the above optimality condition is equivalent to the following stationary condition.

$$0\in\partial f_i(\xi_{r,i})=\lambda\partial|\xi_{r,i}|+y_i\big\{\sigma[r(x_i,a_i^w)-r(x_i,a_i^\ell)+y_i\xi_{r,i}]-1\big\},\tag{64}$$

where $\partial$ denotes partial differential. Noticing that $y_i\in\{-1,1\}$, it can be easily verified that the above equation has unique solution $\xi_{r,i}$ defined by Eq. (15).

# D. Proof of Proposition 2

Note that

$$\xi^{\pi_r}\stackrel{(a)}{=}\xi_{r^{\pi_r}}\stackrel{(b)}{=}\xi_r,\tag{65}$$

where (a) uses Eq. (17) and (b) substitutes Eq. (29) into Eq. (15). Therefore, by using Lemma 3, Eq. (65), and substituting Eq. (29) into Eqs. (8) and (11), we obtain that

$$\mathcal{L}_{N,\lambda}(r^{\pi_r},\xi^{\pi_r})+\eta V_{\beta,\omega}(\pi_{r^\pi},r^{\pi_r})=\mathcal{L}_{N,\lambda}(r,\xi_r)+\eta V_{\beta,\omega}(\pi,r),\tag{66}$$

Since $\Pi_\mathcal{R}\stackrel{\text{def}}{=}\{\pi_r:r\in\mathcal{R}\}$, the following two statements are equivalent.

(P1): $\pi$ is optimal for the offline DPO-COV objective (18), i.e.,

$$\pi\in\arg\min_{\pi'\in\Pi_\mathcal{R}}[\mathcal{L}_{N,\lambda}(r^{\pi'},\xi^{\pi'})+\eta V_{\beta,\omega}(\pi_{r^{\pi'}},r^{\pi'})].$$

(P2): There exists $r\in\arg\min_{r'\in\mathcal{R}}[\mathcal{L}_{N,\lambda}(r^{\pi_{r'}},\xi^{\pi_{r'}})+\eta V_{\beta,\omega}(\pi_{r^{\pi_{r'}}},r^{\pi_{r'}})]$ such that $\pi=\pi_r$.

This along with Eq. (66) implies that (P2) is equivalent to the following statement.

(P3): There exists $r\in\arg\min_{r'\in\mathcal{R}}[\mathcal{L}_{N,\lambda}(r',\xi_{r'})+\eta V_{\beta,\omega}(\pi_{r'},r')]$ such that $\pi=\pi_r$.

By Proposition 1, (P3) is equivalent to the following statement.

(P4): There exist $r\in\mathcal{R}$ and $\xi=\xi_r\in\mathbb{R}^N$ such that $\pi=\pi_r$, and $(\pi,r,\xi)$ is the optimal solution to the offline RLHF-COV objective (12).

So far, we have proved the equivalence among (P1)-(P4), so the first part of this proposition is correct which states that (P1) and (P4) are equivalent.

It remains to prove the second part of this proposition, i.e., to figure out $\xi$ and $r$ given $\pi$ under the assumption that (P1)-(P4) hold. Note that based on the analytical solution (14) of $\pi_r$, $\pi = \pi_r$ required by (P2)-(P4) holds if and only if for any $x \in \mathcal{X}$ there exists $U_\pi(x) \in \mathbb{R}$ such that $r(x, \cdot) = r^\pi(x, \cdot) + U_\pi(x)$. In this case, we have

$$\xi \stackrel{(a)}{=} \xi_r \stackrel{(b)}{=} \xi_{r^\pi} \stackrel{(c)}{=} \xi^\pi,$$

where (a) uses (P4), (b) substitutes $r(x, \cdot) = r^\pi(x, \cdot) + U_\pi(x)$ into Eq. (16), (c) uses $\xi^\pi \stackrel{\text{def}}{=} \xi_{r^\pi}$.

## E. Proof of Proposition 3

The proof logic is exactly the same as that of Proposition 2, with $\eta$ replaced by $-\eta$.

## F. Proof of Theorem 1

Obtain $\widetilde{\pi} \in \arg\min_{\pi \in \Pi_{\mathcal{R}}} \left[ \mathcal{L}_{N,\lambda}(r^\pi, \xi^\pi) + \eta V_{\beta,\omega}(\pi_{r^\pi}, r^\pi) \right]$ by minimizing the offline DPO-COV objective (18). Then based on Proposition (2), there exists $\widetilde{r} \in \mathcal{R}$ such that $(\widetilde{\pi}, \widetilde{r}, \xi^{\widetilde{\pi}})$ ($\xi^{\widetilde{\pi}}$ is defined by Eq. (17)) is the optimal solution to the offline RLHF-COV objective (12), that is,

$$(\widetilde{r}, \xi^{\widetilde{\pi}}) \in \arg\min_{r' \in \mathcal{R}, \xi' \in \mathbb{R}^N} \max_{\pi' \in \Pi} \left[ \mathcal{L}_{N,\lambda}(r', \xi') + \eta V_{\beta,\omega}(\pi', r') \right], \tag{67}$$

$$\widetilde{\pi} = \pi_{\widetilde{r}} \in \arg\max_{\pi' \in \Pi} V_{\beta,\omega}(\pi', \widetilde{r}). \tag{68}$$

Then denote $\widetilde{\pi}_2 \in \arg\max_{\pi' \in \Pi} \min_{r' \in \mathcal{R}, \xi' \in \mathbb{R}^N} \left[ \mathcal{L}_{N,\lambda}(r', \xi') + \eta V_{\beta,\omega}(\pi', r') \right]$ and we have

$$\mathcal{L}_{N,\lambda}(\widetilde{r}, \xi^{\widetilde{\pi}}) + \eta V_{\beta,\omega}(\widetilde{\pi}_2, \widetilde{r})$$

$$\geq \min_{r' \in \mathcal{R}, \xi' \in \mathbb{R}^N} \left[ \mathcal{L}_{N,\lambda}(r', \xi') + \eta V_{\beta,\omega}(\widetilde{\pi}_2, r') \right]$$

$$\stackrel{(a)}{=} \max_{\pi' \in \Pi} \min_{r' \in \mathcal{R}, \xi' \in \mathbb{R}^N} \left[ \mathcal{L}_{N,\lambda}(r', \xi') + \eta V_{\beta,\omega}(\pi', r') \right]$$

$$\stackrel{(b)}{=} \min_{r' \in \mathcal{R}, \xi' \in \mathbb{R}^N} \max_{\pi' \in \Pi} \left[ \mathcal{L}_{N,\lambda}(r', \xi') + \eta V_{\beta,\omega}(\pi', r') \right] \tag{69}$$

$$\stackrel{(c)}{=} \max_{\pi' \in \Pi} \left[ \mathcal{L}_{N,\lambda}(\widetilde{r}, \xi^{\widetilde{\pi}}) + \eta V_{\beta,\omega}(\pi', \widetilde{r}) \right], \tag{70}$$

where (a) uses $\widetilde{\pi}_2 \in \arg\max_{\pi' \in \Pi} \min_{r' \in \mathcal{R}, \xi' \in \mathbb{R}^N} \left[ \mathcal{L}_{N,\lambda}(r', \xi') + \eta V_{\beta,\omega}(\pi', r') \right]$, (b) applies the minimax theorem (Theorem 1 of (Fan, 1953)) to the function $\mathcal{L}_{N,\lambda}(r', \xi') + \eta V_{\beta,\omega}(\pi', r')$ (defined by Eqs. (8) and (11)) which is a concave function of $\pi' \in \Pi$ and a convex function of $(r', \xi') \in \mathcal{R} \times \mathbb{R}^d$, and (c) uses Eq. (67). The above inequality implies that $\widetilde{\pi}_2 \in \max_{\pi' \in \Pi} V_{\beta,\omega}(\pi', \widetilde{r})$ and thus $\widetilde{\pi}_2 = \pi_{\widetilde{r}} \stackrel{(68)}{=} \widetilde{\pi}$. This means

$$\widetilde{\pi} = \widetilde{\pi}_2 \in \arg\max_{\pi' \in \Pi} \min_{r' \in \mathcal{R}, \xi' \in \mathbb{R}^N} \left[ \mathcal{L}_{N,\lambda}(r', \xi') + \eta V_{\beta,\omega}(\pi', r') \right]. \tag{71}$$

Note that for any $\pi \in \Pi$, Eqs. (11), (20) imply that

$$J_{\beta,\omega}(\pi) - J_{\beta,\omega}(\widetilde{\pi}) = V_{\beta,\omega}(\pi) - V_{\beta,\omega}(\widetilde{\pi}). \tag{72}$$

Hence, $\pi_{r^*} \in \arg\max_{\pi \in \Pi} V_{\beta,\omega}(\pi)$ also satisfies

$$\pi_{r^*} \in \arg\max_{\pi \in \Pi} J_{\beta,\omega}(\pi). \tag{73}$$

Finally, we prove the generalization error rate (22) as follows.

$$\max_{\pi \in \Pi} J_{\beta,\omega}(\pi) - J_{\beta,\omega}(\widetilde{\pi})$$

$$\overset{(a)}{=} V_{\beta,\omega}(\pi_{r^*}, r^*) - \eta^{-1} \max_{\pi \in \Pi} \min_{r \in \mathcal{R}, \xi \in \mathbb{R}^N} \big[ \mathcal{L}_{N,\lambda}(r, \xi) + \eta V_{\beta,\omega}(\pi, r) \big]$$

$$+ \eta^{-1} \min_{r \in \mathcal{R}, \xi \in \mathbb{R}^N} \big[ \mathcal{L}_{N,\lambda}(r, \xi) + \eta V_{\beta,\omega}(\widetilde{\pi}, r) \big] - V_{\beta,\omega}(\widetilde{\pi}, r^*)$$

$$\overset{(b)}{\leq} V_{\beta,\omega}(\pi_{r^*}, r^*) - \eta^{-1} \min_{r \in \mathcal{R}} \big[ \mathcal{L}_{N,\lambda}(r, \xi_r) + \eta V_{\beta,\omega}(\pi_{r^*}, r) \big]$$

$$+ \eta^{-1} \big[ \mathcal{L}_{N,\lambda}(r^*, \xi^*) + \eta V_{\beta,\omega}(\widetilde{\pi}, r^*) \big] - V_{\beta,\omega}(\widetilde{\pi}, r^*)$$

$$\overset{(c)}{=} \max_{r \in \mathcal{R}} \Big\{ \mathbb{E}_{x \sim \rho, a \sim \pi_{r^*}(\cdot|x), a' \sim \pi_{\text{base}}(\cdot|x)} \big[ r^*(x,a) - r^*(x,a') - r(x,a) + r(x,a') \big]$$

$$+ \eta^{-1} [\mathcal{L}_{N,\lambda}(r^*, \xi^*) - \mathcal{L}_{N,\lambda}(r, \xi_r)] \Big\}$$

$$\overset{(d)}{\leq} \max_{r \in \mathcal{R}} \Big\{ G_{\mathcal{D}} E_r + \frac{2}{N\eta} \Big[ \|\xi^*\|_1 + \log\Big( \frac{|\mathcal{N}_{1/N}(\mathcal{R})|}{\delta} \Big) \Big] - \frac{E_r^2}{2\eta(3 + e^R)^2} + \frac{7}{N\eta} \Big\}$$

$$\overset{(e)}{\leq} \frac{2}{N\eta} \Big[ \|\xi^*\|_1 + 5\log\Big( \frac{|\mathcal{N}_{1/N}(\mathcal{R})|}{\delta} \Big) \Big] + \frac{\eta G_{\mathcal{D}}^2}{2}(3 + e^R)^2$$

$$\overset{(f)}{\leq} \frac{(G_{\mathcal{D}}^2 + 1)(3 + e^R)}{\sqrt{N}} \sqrt{\|\xi^*\|_1 + 5\log[|\mathcal{N}_{1/N}(\mathcal{R})|/\delta]}, \tag{74}$$

where (a) uses Eqs. (71), (72) and (73), (b) uses $\xi_r \in \arg\min_{\xi \in \mathbb{R}^N} \mathcal{L}_{N,\lambda}(r, \xi)$ as well as $r^* \in \mathcal{R}$ in Assumption 2, (c) uses Eq. (11), (d) uses Assumption 3 and Lemma 8 with $\epsilon = 1/N$ and $E_r :=$ $\sqrt{\mathbb{E}_{\mathcal{D}} |r^*(x_1, a_1^w) - r^*(x_1, a_1^\ell) - r(x_1, a_1^w) + r(x_1, a_1^\ell)|^2}$, (e) uses $1 \leq \log[|\mathcal{N}_{1/N}(\mathcal{R})|/\delta]$ as well as $bE - aE^2 \leq \frac{b^2}{4a}$ for any $a > 0$ and $b, E \in \mathbb{R}$, (f) uses $\eta = \frac{2\sqrt{\|\xi^*\|_1 + 5\log[|\mathcal{N}_{1/N}(\mathcal{R})|/\delta]}}{\sqrt{N}(3 + e^R)}$.

# G. Proof of Theorem 2

The update rule (25) implies that

$$0 \leq t\phi_t(\pi_{r^*}) - t\phi_t(\pi_{t+1})$$

$$\overset{(a)}{=} \sum_{i=1}^{t} \Big\{ \lambda(|\xi_{r^*,i}| - |\xi_i^{\pi_{t+1}}|) + \beta\eta \log \frac{\pi_{r^*}(a_i^{(-1)}|x_i)}{\pi_{t+1}(a_i^{(-1)}|x_i)}$$

$$+ \log \frac{\sigma[r^{\pi_{t+1}}(x_i, a_i^w) - r^{\pi_{t+1}}(x_i, a_i^\ell) + y_i \xi_i^{\pi_{t+1}}]}{\sigma[r^*(x_i, a_i^w) - r^*(x_i, a_i^\ell) + y_i \xi_{r^*,i}]} \Big\}$$

$$\overset{(b)}{\leq} \sum_{i=1}^{t} \Big\{ \lambda(|\xi_i^*| - |\xi_i^{\pi_{t+1}}|) + \beta\eta \log \frac{\pi_{r^*}(a_i^{(-1)}|x_i)}{\pi_{t+1}(a_i^{(-1)}|x_i)}$$

$$+ \log \frac{\sigma[r^{\pi_{t+1}}(x_i, a_i^w) - r^{\pi_{t+1}}(x_i, a_i^\ell) + y_i \xi_i^{\pi_{t+1}}]}{\sigma[r^*(x_i, a_i^w) - r^*(x_i, a_i^\ell) + y_i \xi_i^*]} \Big\}, \tag{75}$$

where (a) uses Eq. (16), $\xi_i^{\pi_{r^*}} = \xi_{r^*,i}$ (by Eq. (65)), and Lemma 2 (with $r$ replaced by $r^*$) and (b) uses the fact that $\xi_{r^*,i} \in \arg\min_{\xi_i \in \mathbb{R}} \{ \lambda|\xi_i| - \log \sigma[r^*(x_i, a_i^w) - r^*(x_i, a_i^\ell) + y_i \xi_i] \}$, the $i$-th component of $\mathcal{L}_{t,\lambda}(r^*, \xi)$ defined in Eq. (8).

Based on Lemmas 9 and 11 (both with $\delta$ replaced by $\delta/2$ and $\pi$ replaced by $\pi_{t+1}$), the following two inequalities hold for $t = 1, \ldots, T$ simultaneously with probability at least $1 - \delta$.

$$\sum_{i=1}^{t} \log \frac{\sigma[r^{\pi_{t+1}}(x_i, a_i^w) - r^{\pi_{t+1}}(x_i, a_i^\ell) + y_i \xi_i^{\pi_{t+1}}]}{\sigma[r^*(x_i, a_i^w) - r^*(x_i, a_i^\ell) + y_i \xi_i^*]}$$

$$\leq 2\log\Big( \frac{2T|\mathcal{N}_\epsilon(\mathcal{R})|}{\delta} \Big) + 4t\epsilon + \sum_{i=1}^{t} \Big\{ \frac{1}{4}|\xi_i^*| + \sigma(R)|\xi_i^{\pi_{t+1}}|$$

$$- \frac{1}{2(3 + e^R)^2} \mathbb{E}_{x \sim \rho, a^{(1)} \sim \pi_i(\cdot|x), a^{(-1)} \sim \pi_{\text{ref}}(\cdot|x)} [f_{\pi_{t+1}}^2(x, a^{(1)}, a^{(-1)})] \Big\}, \tag{76}$$

$$\sum_{i=1}^{t} \log \frac{\pi_{r^*}(a_i^{(-1)}|x_i)}{\pi_{t+1}(a_i^{(-1)}|x_i)} \le \frac{4R}{\beta}\sqrt{2t\log\Big[\frac{2T\mathcal{N}_\epsilon(\mathcal{R})}{\delta}\Big]} + \frac{4t\epsilon}{\beta} + t\mathbb{E}_{x\sim\rho,a\sim\pi_{\mathrm{ref}}(\cdot|x)}\Big[\log\frac{\pi_{r^*}(a|x)}{\pi_{t+1}(a|x)}\Big]. \tag{77}$$

Substituting Eqs. (76) and (77) into Eq. (75), we obtain that

$$
\begin{aligned}
0 \le\;& 4\eta R\sqrt{2t\log\Big[\frac{4T\mathcal{N}_\epsilon(\mathcal{R})}{\delta}\Big]} + 4\eta\epsilon t + \beta\eta t\,\mathbb{E}_{x\sim\rho,a\sim\pi_{\mathrm{ref}}(\cdot|x)}\Big[\log\frac{\pi_{r^*}(a|x)}{\pi_{t+1}(a|x)}\Big] \\
&+ \lambda\sum_{i=1}^{t}(|\xi_i^*| - |\xi_i^{\pi_{t+1}}|) + 2\log\Big(\frac{2T|\mathcal{N}_\epsilon(\mathcal{R})|}{\delta}\Big) + 4t\epsilon + \sum_{i=1}^{t}\Big\{\frac{1}{4}|\xi_i^*| + \sigma(R)|\xi_i^{\pi_{t+1}}| \\
&- \frac{1}{2(3+e^R)^2}\mathbb{E}_{x\sim\rho,a^{(1)}\sim\pi_i(\cdot|x),a^{(-1)}\sim\pi_{\mathrm{ref}}(\cdot|x)}\big[f_{\pi_{t+1}}^2(x,a^{(1)},a^{(-1)})\big]\Big\} \\
\overset{(a)}{\le}\;& 4\eta R\sqrt{2t\log\Big[\frac{4T\mathcal{N}_\epsilon(\mathcal{R})}{\delta}\Big]} + 2\log\Big(\frac{2T|\mathcal{N}_\epsilon(\mathcal{R})|}{\delta}\Big) + 4\eta\epsilon t + 4\epsilon t \\
&- \beta\eta t\,\mathbb{E}_{x\sim\rho,a\sim\pi_{\mathrm{ref}}(\cdot|x)}\Big[\log\frac{\pi_{t+1}(a|x)}{\pi_{r^*}(a|x)}\Big] \\
&+ \sum_{i=1}^{t}\Big\{\frac{5}{4}|\xi_i^*| - \frac{1}{2(3+e^R)^2}\mathbb{E}_{x\sim\rho,a^{(1)}\sim\pi_i(\cdot|x),a^{(-1)}\sim\pi_{\mathrm{ref}}(\cdot|x)}\big[f_{\pi_{t+1}}^2(x,a^{(1)},a^{(-1)})\big]\Big\}, \tag{78}
\end{aligned}
$$

where (a) uses $\lambda\in[\sigma(R),1]$. Then, we have

$$
\begin{aligned}
& J_{\beta,\omega}(\pi_{r^*}) - J_{\beta,\omega}(\pi_{t+1}) \\
\overset{(a)}{=}\;& \mathbb{E}_{x\sim\rho,a\sim\pi_{r^*}(\cdot|x)}\Big[r^*(x,a) - \omega|a| - \beta\log\frac{\pi_{r^*}(a|x)}{\pi_{\mathrm{ref}}(a|x)}\Big] \\
&- \mathbb{E}_{x\sim\rho,a\sim\pi_{t+1}(\cdot|x)}\Big[r^*(x,a) - \omega|a| - \beta\log\frac{\pi_{t+1}(a|x)}{\pi_{\mathrm{ref}}(a|x)}\Big] \\
\overset{(b)}{=}\;& \mathbb{E}_{x\sim\rho,a\sim\pi_{\mathrm{ref}}(\cdot|x)}\Big[r^*(x,a) - \omega|a| - \beta\log\frac{\pi_{r^*}(a|x)}{\pi_{\mathrm{ref}}(a|x)}\Big] \\
&- \mathbb{E}_{x\sim\rho,a\sim\pi_{t+1}(\cdot|x)}\Big[r^*(x,a) - \omega|a| - \beta\log\frac{\pi_{t+1}(a|x)}{\pi_{\mathrm{ref}}(a|x)}\Big] \\
=\;& \beta\mathbb{E}_{x\sim\rho,a\sim\pi_{\mathrm{ref}}(\cdot|x)}\Big[\log\frac{\pi_{t+1}(a|x)}{\pi_{r^*}(a|x)}\Big] + \mathbb{E}_{x\sim\rho,a\sim\pi_{t+1}(\cdot|x)}\Big[\omega|a| + \beta\log\frac{\pi_{t+1}(a|x)}{\pi_{\mathrm{ref}}(a|x)} - r^*(x,a)\Big] \\
&- \mathbb{E}_{x\sim\rho,a\sim\pi_{\mathrm{ref}}(\cdot|x)}\Big[\omega|a| + \beta\log\frac{\pi_{t+1}(a|x)}{\pi_{\mathrm{ref}}(a|x)} - r^*(x,a)\Big] \\
\overset{(c)}{=}\;& \beta\mathbb{E}_{x\sim\rho,a\sim\pi_{\mathrm{ref}}(\cdot|x)}\Big[\log\frac{\pi_{t+1}(a|x)}{\pi_{r^*}(a|x)}\Big] + \mathbb{E}_{x\sim\rho,a\sim\pi_{t+1}(\cdot|x)}\big[r^{\pi_{t+1}}(x,a) - r^*(x,a)\big] \\
&- \mathbb{E}_{x\sim\rho,a\sim\pi_{\mathrm{ref}}(\cdot|x)}\big[r^{\pi_{t+1}}(x,a) - r^*(x,a)\big] \\
\overset{(d)}{=}\;& \beta\mathbb{E}_{x\sim\rho,a\sim\pi_{\mathrm{ref}}(\cdot|x)}\Big[\log\frac{\pi_{t+1}(a|x)}{\pi_{r^*}(a|x)}\Big] - \mathbb{E}_{x\sim\rho,a^{(1)}\sim\pi_{t+1}(\cdot|x),a^{(-1)}\sim\pi_{\mathrm{ref}}(\cdot|x)}\big[f_{\pi_{t+1}}(x,a^{(1)},a^{(-1)})\big] \\
\overset{(e)}{\le}\;& \beta\mathbb{E}_{x\sim\rho,a\sim\pi_{\mathrm{ref}}(\cdot|x)}\Big[\log\frac{\pi_{t+1}(a|x)}{\pi_{r^*}(a|x)}\Big] + \frac{\eta t}{2}(3+e^R)^2 I_t \\
&+ \frac{1}{2\eta t(3+e^R)^2}\Big\{R^2 + \sum_{i=1}^{t}\mathbb{E}_{x\sim\rho,a^{(1)}\sim\pi_i(\cdot|x),a^{(-1)}\sim\pi_{\mathrm{ref}}(\cdot|x)}\big[f_{\pi_{t+1}}^2(x,a^{(1)},a^{(-1)})\big]\Big\} \\
\overset{(f)}{\le}\;& \frac{\eta t}{2}(3+e^R)^2 I_t + \frac{1}{2\eta t} + 4R\sqrt{\frac{2}{t}\log\Big[\frac{4T\mathcal{N}_\epsilon(\mathcal{R})}{\delta}\Big]} + \frac{2}{\eta t}\log\Big(\frac{2T|\mathcal{N}_\epsilon(\mathcal{R})|}{\delta}\Big) \\
&+ 4\epsilon + \frac{4\epsilon}{\eta} + \frac{5}{4\eta t}\sum_{i=1}^{t}|\xi_i^*|,
\end{aligned}
$$

where (a) uses Eq. (20), (b) uses Eq. (14) which implies that $r^*(x,a) - \omega|a| - \beta \log \frac{\pi_{r^*}(a|x)}{\pi_{\text{ref}}(a|x)} = \beta \log Z_{r^*}(x)$ does not rely on $a$, (c) uses Eqs. (16), (d) uses Eq. (46), (e) applies Cauchy-Schwartz inequality to Eq. (56), (f) uses Eq. (78) and $3 + e^R > R > 0$. Finally, we conclude the proof by averaging the above inequality over $t \in \{1, 2, \ldots, T\}$ as follows.

$$\mathbb{E}\big[J_{\beta,\omega}(\pi_{r^*}) - J_{\beta,\omega}(\pi_{\widehat{T}})\big] = \frac{1}{T} \sum_{t=1}^{T} \big[J_{\beta,\omega}(\pi_{r^*}) - J_{\beta,\omega}(\pi_{t+1})\big]$$

$$\stackrel{(a)}{\leq} 6\eta G_{\text{on}}(3 + e^R)^2 \log(T+2) + \frac{3 \log T}{2\eta T} + 8R \sqrt{\frac{2}{T} \log\Big[\frac{4T \mathcal{N}_\epsilon(\mathcal{R})}{\delta}\Big]}$$

$$+ \frac{6 \log T}{T\eta} \log\Big(\frac{2T|\mathcal{N}_\epsilon(\mathcal{R})|}{\delta}\Big) + 4\epsilon + \frac{4\epsilon}{\eta} + \frac{15 \log T}{4T\eta} \sum_{i=1}^{T} |\xi_i^*|$$

$$\stackrel{(b)}{\leq} 6(3 + e^R) \log(T+2) \sqrt{\frac{G_{\text{on}}}{T} \Big[\log\Big(\frac{4T|\mathcal{N}_{1/T}(\mathcal{R})|}{\delta}\Big) + \|\xi^*\|_1\Big]} + \frac{3(3 + e^R)(\log T)\sqrt{G_{\text{on}}}}{2\sqrt{T \log[2T\mathcal{N}_{1/T}(\mathcal{R})/\delta]}}$$

$$+ 8R \sqrt{\frac{2}{T} \log\Big[\frac{4T\mathcal{N}_{1/T}(\mathcal{R})}{\delta}\Big]} + 6(3 + e^R)(\log T) \sqrt{\frac{G_{\text{on}}}{T} \log\Big(\frac{4T|\mathcal{N}_{1/T}(\mathcal{R})|}{\delta}\Big)}$$

$$+ \frac{4}{T} + 4(3 + e^R) \sqrt{\frac{G_{\text{on}}}{T \log[2T\mathcal{N}_{1/T}(\mathcal{R})/\delta]}} + \frac{15(3 + e^R)(\log T)\sqrt{G_{\text{on}}}}{4\sqrt{T \log 42T\mathcal{N}_{1/T}(\mathcal{R})/\delta] + T\|\xi^*\|_1}} \|\xi^*\|_1$$

$$\stackrel{(c)}{\leq} (6 + 1.5 + 8\sqrt{2} + 6 + 4 + 4)(3 + e^R)(\log T) \sqrt{\frac{G_{\text{on}}}{T} \Big[\log\Big(\frac{4T|\mathcal{N}_{1/T}(\mathcal{R})|}{\delta}\Big) + \|\xi^*\|_1\Big]}$$

$$+ \frac{15(3 + e^R)(\log T)\sqrt{G_{\text{on}}}}{4\sqrt{T \log[4T\mathcal{N}_{1/T}(\mathcal{R})/\delta] + T\|\xi^*\|_1}} \big\{ \log[4T\mathcal{N}_{1/T}(\mathcal{R})/\delta] + \|\xi^*\|_1 \big\}$$

$$\leq 37(3 + e^R)(\log T) \sqrt{\frac{G_{\text{on}}}{T} \Big[\log\Big(\frac{4T|\mathcal{N}_{1/T}(\mathcal{R})|}{\delta}\Big) + \|\xi^*\|_1\Big]},$$

where (a) uses $\sum_{t=1}^{T} \frac{1}{t} \leq 1 + \log T \leq 3 \log T$, $\sum_{t=1}^{T} \frac{1}{\sqrt{t}} \leq 2\sqrt{T}$ and Eq. (57), (b) uses $\eta = \frac{\sqrt{\log[4T\mathcal{N}_{1/T}(\mathcal{R})/\delta] + \|\xi^*\|_1}}{(3 + e^R)\sqrt{TG_{\text{on}}}}$, $\epsilon = \frac{1}{T}$, and (c) uses $G_{\text{on}} \geq 1$ (by Eq. (26)), $R < 3 + e^R$, $\log(T+2) \leq 2 \log T$ and $\log\big(\frac{4T|\mathcal{N}_{1/T}(\mathcal{R})|}{\delta}\big) \geq \log T \geq 1$.

