# OpenReview forum: "Provably Mitigating Corruption, Overoptimization, and Verbosity Simultaneously in Offline and Online RLHF/DPO Alignment"
_ICML.cc/2025/Conference — Submitted to ICML 2025_

### Official Review · Reviewer_j1Ec · 2025-03-02

**Overall Recommendation:** 3

**Summary:**

Reinforcement learning from human feedback (RLHF) aims to align generative models with human preferences. However, the quality of alignment training can be compromised by corrupted preferences, reward overoptimization, and bias toward verbose outputs. This paper proposes RLHF-COV and DPO-COV, two algorithms designed to simultaneously address these issues in both offline and online settings.

**Claims And Evidence:**

For each of the three issues, i.e., preference corruption, reward overoptimization, and output verbosity, the vanilla RLHF objective is augmented with a corresponding loss component to address the issue. To account for potential corruption in preference, a noise term is added to the loss based on the Bradley-Terry model. To address reward overoptimization, a "pessimistic MLE" is adopted where a baseline policy is used to reduce overestimation of low-quality out-of-distribution samples. Lastly, to control verbosity, a length-based penalty is incorporated into the objective.

Each of these loss components, or similar variants, is conceptually sound and has been studied in prior works, albeit in isolation. This paper naturally integrates them into a single objective. While the proposed algorithms are well-motivated, the experimental evaluation remains limited. The study explores only a narrow set of tasks, primarily math and reasoning, with minimal analysis. Key questions remain: How does the algorithm perform on a broader range of tasks? How much corruption exists in the preference data, and to what extent can the proposed approach mitigate it? To what degree is reward overoptimization a concern, and why is tuning the beta parameter in vanilla DPO insufficient to address it?

**Essential References Not Discussed:**

None that I am aware of.

**Experimental Designs Or Analyses:**

The presented experimental results and analysis are sound, but to fully assess how well the proposed algorithms address the three issues in existing alignment methods, additional experimental results should be included. Please see above.

**Methods And Evaluation Criteria:**

The proposed methods make sense for addressing the issues of corrupted preferences, reward overoptimization, and length biases.

For evaluation, the paper uses length-controlled win rates in AlpacaEval 2.0 and task performance on math and reasoning benchmarks, including GSM8K and ARC. While the evaluation setup is reasonable, the experimental results and analysis are minimal, making it difficult to fully assess the extent to which the identified issues are present in the setup and how effectively the proposed algorithms and baselines address them. Please see above.

**Other Comments Or Suggestions:**

None.

**Other Strengths And Weaknesses:**

None.

**Questions For Authors:**

Q. How much more effective is the proposed pessimistic MLE compared to the regular KL-penalized objective in reducing overoptimization? What baseline policy is used in the experiments?

**Relation To Broader Scientific Literature:**

The issues of preference corruption, reward overoptimization, and length biases are important challenges in alignment, typically studied in isolation. While more experiments are needed to assess whether addressing all of these issues in a single optimization is the most effective -- especially given that the extent to which each issue is present can vary -- the paper introduces algorithms to tackle them simultaneously and provides theoretical results on generalization error.

**Theoretical Claims:**

Theoretical claims regarding the generalization error rate of the proposed algorithms have been reviewed at a high level but not to the extent of verifying the constants in the bounds.

---

> ### Author Rebuttal · Authors · 2025-04-01
>
> **Claims And Evidence (1):** The study explores only a narrow set of tasks, primarily math and reasoning, with minimal analysis. Key questions remain: How does the algorithm perform on a broader range of tasks?
>
> **A:** Thanks for your question. We are conducting experiments on the new tasks.
>
> **Claims And Evidence (2):** How much corruption exists in the preference data, and to what extent can the proposed approach mitigate it?
>
> **A:** Thanks for your questions. It is hard to count the number of corrupted labels in the large datasets, but we are sure there are corruptions in the AI-generated labels, so we use the performance gap (by LC win-rates and accuracies) between our algorithm and the 3 algorithms that do not directly tackle corruption (i.e., pessimistic/optimistic DPO algorithm, length-regularized DPO algorithm and vanilla DPO algorithm) to roughly reflect the extent of corruption mitigation.
>
> However, we are working on the new experiments by corrupting x\% of the labels in the preference data with various x.
>
> **Claims And Evidence (3):** To what degree is reward overoptimization a concern, and why is tuning the beta parameter in vanilla DPO insufficient to address it?
>
> **A:** Thanks for your questions. Note that the robust DPO algorithm (corruption-only), length-regularized DPO algorithm (verbosity-only) and vanilla DPO algorithm in our experiments do not directly deal with over-optimization. Tables 1 and 3 show that the LC-win rates of these 3 algorithms are 0.57\%-1.68\% lower than those of our DPO-COV algorithms. Table 2 shows that these 3 algorithms are 0.38\%-3.16\% less accurate than our DPO-COV algorithm. These performance gaps show the effect of overoptimization.
>
> Tuning beta alone can still cause overoptimization, because the KL-regularization could only control the scale of the gradient per training example, while adding the pessimistic term can further modify the gradient direction [1].
>
> [1] Liu, Z., Lu, M., Zhang, S., Liu, B., Guo, H., Yang, Y., ... \& Wang, Z. Provably Mitigating Overoptimization in RLHF: Your SFT Loss is Implicitly an Adversarial Regularizer. In The Thirty-eighth Annual Conference on Neural Information Processing Systems.
>
> **Questions For Authors:** How much more effective is the proposed pessimistic MLE compared to the regular KL-penalized objective in reducing overoptimization? What baseline policy is used in the experiments?
>
> **A:** Thanks for your questions. The pessimistic MLE is proposed not by us but by (Liu et al., 2024c; Cen et al., 2024; Ji et al., 2024; Yang et al., 2024), which have empirically demonstrated that the pessimistic MLE significantly outperforms the regular KL-penalized objective (i.e., vanilla DPO) in reducing overoptimization. For example, in Table 1 of (Liu et al., 2024c), the win rate of pessimistic MLE VS vanilla DPO is 56\%. As to the baseline policy $\pi_{\rm base}$, Line 307 left side said "Algorithm 1 takes $\pi_{\rm base}(\cdot|x)$ as the distribution of the preferable responses $a_i^w$ given $x_i=x$, which is well covered by $\mathcal{D}$."

---

### Official Review · Reviewer_2BVC · 2025-03-12

**Overall Recommendation:** 2

**Summary:**

The paper studied corruption, overoptimization, and verbosity simultaneously in offline and offline LLM alignment problems (RLHF and DPO). The authors give both theoretical and empirical guarantees.

**Claims And Evidence:**

The claims are mostly correct.

**Essential References Not Discussed:**

The paper missed the important related work [1].

[1].Chowdhury, Sayak Ray, Anush Kini, and Nagarajan Natarajan. "Provably robust dpo: Aligning language models with noisy feedback." arXiv preprint arXiv:2403.00409 (2024).

**Experimental Designs Or Analyses:**

The experiments are limited, which does not quite demonstrate the overall effectiveness of the proposed algorithm since the difference of win rates in Table 1 is subtle.

**Methods And Evaluation Criteria:**

They make sense.

**Other Comments Or Suggestions:**

The equivalence of our proposed RLHF-COV and DPO-COV algorithms is incremental since DPO is a direct preference version for RLHF. And the equivalence of DPO and RLHF has been proved in previous work.

**Other Strengths And Weaknesses:**

Strengths：
The writing is clear, and I like that they give both theoretical and empirical results.


Weaknesses：
1. The paper missed some important related work. See the Questions part for more details.
2. The motivation to tackle the three problems in LLM alignment is unclear.
3. The paper seems to be a combination of the techniques from the three areas.
4. There are too many hyperparameters to be set, which is unrealistic in the real world.

**Questions For Authors:**

1.  Please discuss and compare your results with [1] for robustness.
2.  What is the motivation to tackle the corruption, overoptimization, and verbosity simultaneously in the real world?
3. What is the reason for designing $y_i\xi_i$ instead of  $\xi_i$ in loss function in Lines 158-160.
4. The assumption 3 is all-policy coverage. Can your method improve to singl-policy coverage?


[1].Chowdhury, Sayak Ray, Anush Kini, and Nagarajan Natarajan. "Provably robust dpo: Aligning language models with noisy feedback." arXiv preprint arXiv:2403.00409 (2024).

**Relation To Broader Scientific Literature:**

The problem studied in this paper is interesting.

**Theoretical Claims:**

The theoretical claims are mostly correct.

---

> ### Author Rebuttal · Authors · 2025-04-01
>
> **Questions For Authors (1):** Please discuss and compare your results with [1] for robustness.
>
> [1] Chowdhury, Sayak Ray, Anush Kini, and Nagarajan Natarajan. "Provably robust DPO: Aligning language models with noisy feedback." ArXiv:2403.00409 (2024).
>
> **A:** Thanks for bringing this important work to our attention. [1] proposes a robust DPO method that solves data corruption well by incorporating a certain fixed label flipping probability $\epsilon\in(0,1)$ into the likelihood function, but it does not explicitly solve over-optimization and verbosity, while our work solves all these three issues simultaneoulsly. As to theoretical results, both works achieve generalization error rates of $\mathcal{O}(1/\sqrt{n})$ with $n$ possibly corrupted samples, but [1] uses expected true reward as the generalization error measure, while our generalization error measure (Eq. (20)) strikes a trade-off among expected true reward, distance to the reference policy (to solve over-optimization) and length penalty (to solve verbosity). The experimental results are not directly comparable as they involve different tasks and datasets. We will cite [1] in our revised work.
>
> **Questions For Authors (2):** What is the motivation to tackle the corruption, overoptimization, and verbosity simultaneously in the real world?
>
> **A:** Good question. All these three issues exist and can yield undesirable consequences in the real world as will be elaborated below. That motivates us to tackle these issues simultaneously.
>
> The "Corruption" paragraph in the introduction reveals the reason and dangers of corruption in the real world: "However, preference labels given by human may be corrupted due to inexperience, inattention, personal bias, unclear context, and even malicious falsification (Bukharin et al., 2024). For instance, when fine-tuning LLM for automated content moderation on social media, malicious annotators may mislabel harmful contents like misinformation and hate speech as preferable, which misleads the LLM to generate such harmful contents." Hence, we need to tackle corruption.
>
> The "Overoptimization" paragraph in the introduction reveals the reason and undesirable consequence of overoptimization: "RLHF and DPO may overoptimize the reward model, yielding LLM responses of high estimated reward but low actual quality (Gao et al., 2023; Casper et al., 2023)." This contradicts the goal of RLHF and DPO to make LLM responses helpful, honest, and harmless. Hence, we need to tackle overoptimization.
>
> The "Verbosity" paragraph in the introduction said "LLM aligned by vanilla RLHF and DPO is likely to prefer verbose but possibly low-quality responses". In the real world, verbose responses can waste users' time and thus lose users. This consequence will be added to the revision. Hence, we need to tackle verbosity.
>
> **Questions For Authors (3):** What is the reason for designing $y_i\xi_i$ instead of $\xi_i$ in loss function in Lines 158-160?
>
> **A:** Good question. As shown in the footnote in the page 3, using $y_i\xi_i$ ensures that $\mathbb{P}(y_i|a_i^{(1)},a_i^{(-1)})=\sigma[r^*(x_i,a_i^{w})-r^*(x_i,a_i^{\ell})+y_i\xi_i^*], y_i\in\{-1,1\}$ is a valid probability measure satisfying $\sum_{y_i\in\{-1,1\}}\mathbb{P}(y_i|a_i^{(1)},a_i^{(-1)})=1$. To elaborate, since $y_i=1$ means $a_i^{w}=a_i^{(1)}$ and $a_i^{\ell}=a_i^{(-1)}$, we have $\mathbb{P}(y_i=1|a_i^{(1)},a_i^{(-1)})=\sigma[r^*(x_i,a_i^{(1)})-r^*(x_i,a_i^{(-1)})+\xi_i^*]$. Similarly, since $y_i=-1$ means $a_i^{w}=a_i^{(-1)}$ and $a_i^{\ell}=a_i^{(1)}$, we have $\mathbb{P}(y_i=1|a_i^{(1)},a_i^{(-1)})=\sigma[r^*(x_i,a_i^{(-1)})-r^*(x_i,a_i^{(1)})-\xi_i^*]=1-\sigma[r^*(x_i,a_i^{(1)})-r^*(x_i,a_i^{(-1)})+\xi_i^*]$, so $\sum_{y_i\in\{-1,1\}}\mathbb{P}(y_i|a_i^{(1)},a_i^{(-1)})=1$.
>
> **Questions For Authors (4):** The assumption 3 is all-policy coverage. Can your method improve to single-policy coverage?
>
> **A:** Assumption 3 only involve two fixed policies, the baseline policy $\pi_{\rm base}$ and the true optimal policy $\pi_{r^*}$. Why does the reviewer think it is all-policy coverage? Actually our Assumption 3 is the same as Assumption 2 of [1], which is claimed by [1] as single-policy coverage.
>
> [1] Ji, X., Kulkarni, S., Wang, M., and Xie, T. (2024). Self-play with adversarial critic: Provable and scalable offline alignment for language models. ArXiv:2406.04274.

---

### Official Review · Reviewer_x9cp · 2025-03-14

**Overall Recommendation:** 2

**Summary:**

The authors identify three key challenges in LLM alignment: corruption, overoptimization, and verbosity. To address these issues holistically, they propose a unified approach through generalized formulations of RLHF and DPO called RLHF-COV and DPO-COV, respectively.

These formulations incorporate: noise modeling to mitigate corruption, optimistic/pessimistic regularizers to prevent overoptimization, and a length penalty to discourage verbosity. The authors present both offline and online versions of these objectives.

The authors provide win rates against GPT-4 on AlpacaEval and improvements on math and reasoning datasets to demonstrate the impact of the various improvements in their paper in the offline DPO-COV setting and have online results in the appendix.

**Claims And Evidence:**

The main claim of this paper is that the they develop a unified algorithm to simultaneously address corruption, overoptimization, and verbosity issues in LLM alignment. This claim is supported through the extensive theoretical formulations and empirical results in the paper.

The only claim I find a bit problematic is the verbosity reducing one. The main issue is that verbosity reduction is addressed through a universal length penalty which would not take into account prompt-specific length requirements. The results don't include numbers on average response lengths, so it is unclear how well the length regularizer works. Another concern is that reducing length might affect reasoning performance because of shortened reasoning chains, but the evals on reasoning and math benchmarks don't sustain this.

**Essential References Not Discussed:**

N/A

**Experimental Designs Or Analyses:**

## Experiment suggestions
### Generalizability
It is hard to judge the performance of DPO-COV with just win rates against GPT-4 on AlpacaEval. It would be nice to see a few more benchmarks (ArenaHard and MT-Bench) or different base models (llama3-8b).

### Effect of Corruption levels in data
It would be helpful to see how different levels of corruption in the preference data affects alignment. The authors speak about malicious actors deliberately corrupting preference data as a motivation, so it would be helpful to see how robust this method is to an issue like that.

### Overoptimizations
An analysis into the patterns of reward hacking that were observed in the vanilla DPO run vs. DPO-COV run would also help illustrate how all these techniques work together to make a better model.

**Methods And Evaluation Criteria:**

The base model and dataset choices are all reasonable. However, evaluations are a bit lacking. More details under experiments and analysis.

**Other Comments Or Suggestions:**

N/A

**Other Strengths And Weaknesses:**

All 3 mitigation strategies are proposed in previous works as cited in the paper. The main contribution of this paper is developing an objective that combines all three together. This paper also doesn't have adequate experiments to empirically study the effect of all three objectives applied together. Therefore, the novelty of this paper is limited.

**Questions For Authors:**

N/A

**Relation To Broader Scientific Literature:**

The paper proposed a unified approach to addressing very common issues in LLM alignment. Mitigating and finding good solutions to these problems is a priority in the broader research community. This paper shows that noise modeling to mitigate corruption, optimistic/pessimistic regularizers to prevent overoptimization, and a length penalty to discourage verbosity can all work together to create a better aligned model.

**Theoretical Claims:**

This paper is quite dense in terms of theoretical proofs and claims. I checked all the RLHF-COV proofs and found no issues.

---

> ### Author Rebuttal · Authors · 2025-04-01
>
> **Claims And Evidence:** The only claim I find a bit problematic is the verbosity reducing one. The main issue is that verbosity reduction is addressed through a universal length penalty which would not take into account prompt-specific length requirements. The results don't include numbers on average response lengths, so it is unclear how well the length regularizer works. Another concern is that reducing length might affect reasoning performance because of shortened reasoning chains, but the evals on reasoning and math benchmarks don't sustain this.
>
> **A:** Prompt-specific length is a good future direction, which may be tackled by using length penalty coefficient $\omega(x)$ that depends on the prompt $x$. The larger length required for $x$, the smaller $\omega(x)$ should be.
>
> We are working on experiments to include the numbers on average response lengths. Thanks for your suggestion.
>
> For reasoning, we could use small length penalty coefficient to guarantee intact necessary reasoning steps, while trimming unnecessary steps and tokens. In the future, we may apply different levels of length penalty to different reasoning steps, to ensure a concise description of each step, rather than to reduce the number of steps.
>
> **Generalizability:** It is hard to judge the performance of DPO-COV with just win rates against GPT-4 on AlpacaEval. It would be nice to see a few more benchmarks (ArenaHard and MT-Bench) or different base models (llama3-8b).
>
> **A:** Thanks for your suggestion. We are working on the new experiments.
>
> **Effect of Corruption levels in data:** It would be helpful to see how different levels of corruption in the preference data affects alignment. The authors speak about malicious actors deliberately corrupting preference data as a motivation, so it would be helpful to see how robust this method is to an issue like that.
>
> **A:** Thanks for your suggestion. We are working on the new experiments by corrupting x\% of the labels in the preference data with various x.
>
> **Overoptimizations:** An analysis into the patterns of reward hacking that were observed in the vanilla DPO run vs. DPO-COV run would also help illustrate how all these techniques work together to make a better model.
>
> **A:** Thanks for your suggestion. We are working on the new experiments.

---

### Official Review · Reviewer_8ciG · 2025-03-17

**Overall Recommendation:** 2

**Summary:**

This paper introduces RLHF-COV and DPO-COV to mitigate performance degradation caused by corrupted preferences, reward overoptimization, and bias toward verbosity.
To this end, the authors apply three techniques: a noise regularizer to enhance robustness against corrupted preferences, pessimistic MLE to handle reward over-optimization, and a length penalty to control the verbosity issue simultaneously.

The authors first derive the offline RLHF-COV objective by incorporating all techniques into the RLHF objective. Then, they derive the offline DPO-COV objective by solving the offline RLHF-COV objective analytically, as proved in Proposition 1.
Theoretically, the authors prove that RLHF-COV and DPO-COV are equivalent (Proposition 2).
Moreover, with a bounded reward and a sufficiently large dataset, they show that the performance gap between the solution of offline DPO-COV and the solution of the desirable objective (20) is bounded with high probability.

Additionally, the authors provide the objectives for online RLHF-COV and online DPO-COV.
Finally, the authors demonstrate that offline DPO-COV achieves promising performance on the Argilla-DPO-Mix-7K dataset.

**Claims And Evidence:**

Yes

**Essential References Not Discussed:**

-

**Experimental Designs Or Analyses:**

The experiments should be revised.

First, to show the LC-win rate, Argilla-DPO-Mix-7K is used, while performance is compared using reasoning tasks such as GSM8K and ARC.
Why are different tasks required?

In addition, in the experiments section, readers may expect that DPO-COV is empirically robust against corrupted preferences and reward overoptimization.
However, why does the performance on reasoning tasks reveal such desirable properties of DPO-COV?

Finally, among the proposed methods, only the performance of offline DPO-COV is reported.

**Methods And Evaluation Criteria:**

Need more explanation.

**Other Comments Or Suggestions:**

The equations have poor readability.
For example, in Equation (12), (18), (23), and (24), what is the role of $\lbrace$ and $\rbrace$?
In addition, the line breaks in the equations are very hard to follow.

**Other Strengths And Weaknesses:**

-

**Questions For Authors:**

**[Question about Analysis]**

I am confused by the gap between the theory and my intuition.

I do not see any assumptions about noise; therefore, I may assume very large noise.
In this case, preferences become random, making it impossible to guarantee the performance of any RLHF algorithm, including DPO-COV, since the given data contains no information about preferences.

However, Theorem 1 guarantees the performance gap, which suggests that DPO-COV can obtain a reasonable solution.
Could you clarify this confusion?

**[Question about objective (20)]**

The authors suggest that objective (20) is a desirable objective.
However, I am not sure that (20) is truly a desirable goal, as there are many ways to address the verbosity issue that are not aligned with objective (20) (For example, SimPO or GRPO, etc)

**Relation To Broader Scientific Literature:**

Handling real-world preference datasets that include corrupted preferences.

**Theoretical Claims:**

Yes

---

> ### Author Rebuttal · Authors · 2025-04-01
>
> **Experimental Designs Or Analyses (1):** To show the LC-win rate, Argilla-DPO-Mix-7K is used, while performance is compared using reasoning tasks such as GSM8K and ARC. Why are different tasks required?
>
> **A:** Argilla-DPO-Mix-7K is the preference dataset we use for training, while GSM8K and ARC are benchmarks used for evaluation.
>
> **Experimental Designs Or Analyses (2):** In addition, in the experiments section, readers may expect that DPO-COV is empirically robust against corrupted preferences and reward overoptimization. However, why does the performance on reasoning tasks reveal such desirable properties of DPO-COV?
>
> **A:** Math and reasoning are two important categories of tasks in LLM evaluation [1], the superior performance of DPO-COV over these benchmarks compared to vanilla DPO exactly show that our method is robust against corrupted preferences and reward overoptimization.
>
> [1]. Gao, L. et. al. (2024a). A framework for few-shot language model evaluation. \url{https://zenodo.org/records/12608602}.
>
> **Experimental Designs Or Analyses (3):** Among the proposed methods, only the performance of offline DPO-COV is reported.
>
> **A:** The performance of online methods is reported in Appendix A due to limited space of the main text.
>
> **Other Comments Or Suggestions:**
>
> The equations have poor readability. For example, in Equation (12), (18), (23), and (24), what is the role of
> $\{$ and $\}$? In addition, the line breaks in the equations are very hard to follow.
>
> **A:** Thanks for your suggestions. We will simplify these equations respectively as follows.
> $$\min _ {r\in\mathcal{R},\xi\in\mathbb{R}^N} [\max _ {\pi\in\Pi} \mathcal{L} _ {N,\lambda}(r,\xi)+\eta V _ {\beta,\omega}(\pi,r)],$$
>
> $$\min _ {\pi\in\Pi _ {\mathcal{R}}} [\mathcal{L} _ {N,\lambda}(r^{\pi},\xi^{\pi})+\eta V _ {\beta,\omega}(\pi _ {r^{\pi}},r^{\pi})],$$
>
> $$\pi_{t+1}\in{\arg\min} _ {\pi\in\Pi}\min _ {r\in\mathcal{R},\xi^{(t)}\in\mathbb{R}^t}[\mathcal{L} _ {t,\lambda}(r,\xi^{(t)})-\eta V _ {\beta,\omega}(\pi,r)],$$
>
> $$\pi _ {t+1} \in {\arg\min} _ {\pi\in\Pi _ {\mathcal{R}}} [\mathcal{L} _ {t,\lambda}(r^{\pi},\xi^{\pi,(t)})-\eta V _ {\beta,\omega}(\pi_{r^{\pi}},r^{\pi})].$$
>
> The {   } in the original equations unnecessarily repeat the expressions of the functions $\mathcal{L} _ {N,\lambda}$ (similarly $\mathcal{L} _ {t,\lambda}$) and $V _ {\beta,\omega}$ above that have already been defined in Eqs. (8) and (10) respectively, so we removed such repetitions.
>
> **Question about Analysis:** I do not see any assumptions about noise; therefore, I may assume very large noise. In this case, preferences become random, making it impossible to guarantee the performance of any RLHF algorithm, including DPO-COV, since the given data contains no information about preferences. However, Theorem 1 guarantees the performance gap, which suggests that DPO-COV can obtain a reasonable solution. Could you clarify this confusion?
>
> **A:** The convergence rate in Theorem 1 is $\mathcal{O}(\sqrt{||\xi^*|| _ 1/N})$ where $||\xi^*|| _ 1=\sum _ {i=1}^N|\xi_i^*|$ is the norm of the true noise. Right after Theorem 1, we implicitly assume the upper bound on the true noise by saying "Hence, as long as $||\xi^*|| _ 1\le\mathcal{O}[\log(N)]$ (much weaker than Assumption 4.2 of (Bukharin et al., 2024) that there exist constants $c_0,c _ {\infty}>0$ such that $\xi^*$ has at most $c_0$ nonzero entries and they range in $[-c_{\infty},c_{\infty}]$), the generalization error rate (22) has the order of $\mathcal{O}[\log(N)/\sqrt{N}]$."
>
> **Question about objective (20):** The authors suggest that objective (20) is a desirable objective. However, I am not sure that (20) is truly a desirable goal, as there are many ways to address the verbosity issue that are not aligned with objective (20) (For example, SimPO or GRPO, etc.)
>
> **A:** I agree that there are many methods to address the verbosity issue, and also there can be many generalization measures (including our Eq. (20)) that accounts for verbosity. All these methods and measures align in the final goal to strike a trade-off between expected reward and length. Therefore, as long as the length penalty coefficient controlling such trade-off is proper for a specific task, Eq. (20) is a desirable goal.

---

> > ### Comment · Reviewer_8ciG · 2025-04-05
> >
> > First of all, I have an additional question:
> > - There are many RLHF algorithms for reasoning tasks, such as GRPO, RLOO, and REINFORCE++. Could you compare the proposed algorithm (possibly RLHF-COV) with these methods?
> > In addition, only one dataset and base model are used in the experiments - more datasets and base models should be included, as other reviewers have pointed out.
> >
> > In addition, I'd like to clarify the questions:
> >
> > 1. Argilla-DPO-Mix7K contains test dataset. However, the authors use different datasets to evaluate it. Why?
> > 2. I agree that math and reasoning tasks are very important tasks. However, superior performance on these tasks does not directly indicate robustness against corruption or reward overoptimization. To support this claim, you should demonstrate that the dataset is indeed corrupted, or that the model trained by DPO is overoptimized, whereas the model trained by DPO-COV is not.
> > 3.I mean, why are there no results for RLHF-COV?

---

> > > ### Author Response · Authors · 2025-04-08
> > >
> > > **Additional question:** There are many RLHF algorithms for reasoning tasks, such as GRPO, RLOO, and REINFORCE++. Could you compare the proposed algorithm (possibly RLHF-COV) with these methods? In addition, only one dataset and base model are used in the experiments - more datasets and base models should be included, as other reviewers have pointed out.
> > >
> > > **A:** We are conducting these new experiments as you suggested.
> > >
> > > **Question 1:** Argilla-DPO-Mix7K contains test dataset. However, the authors use different datasets to evaluate it. Why?
> > >
> > > **A:** In the updated Table 1: https://docs.google.com/document/d/1c_A6F5_VWR0VGLLUbKCaTko9WrvGsx1g39ovYQJnqUk/edit?tab=t.0, we have compared the negative log likelihood loss of the chosen responses among the offline DPO-type algorithms, over a hold-out test set of the Argilla-DPO-Mix7K dataset, which shows that our algorithm achieves the lowest test loss.
> > >
> > > **Question 2:** I agree that math and reasoning tasks are very important tasks. However, superior performance on these tasks does not directly indicate robustness against corruption or reward overoptimization. To support this claim, you should demonstrate that the dataset is indeed corrupted, or that the model trained by DPO is overoptimized, whereas the model trained by DPO-COV is not.
> > >
> > > **A:** We have randomly corrupted 25\% labels of the Argilla data. The results on this corrupted Argilla data are shown in Table 2 of https://docs.google.com/document/d/1c_A6F5_VWR0VGLLUbKCaTko9WrvGsx1g39ovYQJnqUk/edit?tab=t.0, which indicates that both our DPO-COV and the robust DPO are more robust to the corruption than the other non-robust DPO variants. We are working on the new experiments that compare the overoptimization level.
> > >
> > > **Question 3:** Why are there no results for RLHF-COV?
> > >
> > > **A:** RLHF-type algorithms such as RLHF and RLHF-COV are theoretically equivalent to their DPO counterparts (e.g. Our Propositions 2 and 3), but require additional training of large reward models. Therefore, like many other works that propose a new DPO-type algorithm, our experiments focus on DPO-type algorithms, while RLHF-COV is used as an intermediate step to derive our DPO-COV algorithms.

---

### Decision · Program_Chairs · 2025-05-01

**Decision:**

Reject

**Comment:**

This paper introduces RLHF-COV and DPO-COV, two algorithms that simultaneously address three challenges in LLM alignment: corrupted preferences, reward overoptimization, and verbosity. These methods are theoretically motivated and support both offline and online learning settings. The authors derive formal equivalence between the RLHF and DPO variants and prove generalization guarantees under corrupted feedback. While the theoretical contributions are sound, reviewers found the empirical evaluation limited and were unconvinced by the claims of robustness and generalization based on the presented experiments.

Despite its theoretical rigor, the paper suffers from limited empirical validation, with most experiments restricted to a single dataset and model. Several reviewers noted the absence of comparisons with key baselines such as GRPO, REINFORCE++, and modern RLHF/DPO variants on reasoning tasks. Claims of robustness are not directly tested, and while new experiments with artificially corrupted labels were added post-rebuttal, concerns remain about their realism and the limited scope. The generalization of results to diverse models and datasets is unclear. Reviewers also found the verbosity mitigation results inconclusive, as no response length statistics were reported. Overall, the current evaluation falls short in validating its practical utility. Key concerns around generalization, baseline comparisons, and realism of the corruption model remain unresolved.

Based on the common consensus, therefore, the authors are recommended to consider submitting to the next suitable venue addressing all the concerns once all the necessary modifications are incorporated.